# RANDOMIZED SHARPNESS-AWARE TRAINING FOR BOOSTING COMPUTATIONAL EFFICIENCY IN DEEP LEARNING

## ABSTRACT

By driving optimizers to converge to flat minima, sharpness-aware learning algorithms (such as SAM) have shown the power to achieve state-of-art performances. However, these algorithms will generally incur one extra forward-backward propagation at each training iteration, which largely burdens the computation especially for scalable models. To this end, we propose an efficient training scheme, called Randomized Sharpness-Aware Training (RST). Optimizers in RST would perform a Bernoulli trial at each iteration to choose randomly from base algorithms (SGD) and sharpness-aware algorithms (SAM) with a probability arranged by a predefined scheduling function. Due to the mixture of base algorithms, the overall count of propagation pairs could be largely reduced. Also, we give theoretical analysis on the convergence of RST. Then, we empirically study the computation cost and effect of various types of scheduling functions, and give directions on setting appropriate scheduling functions. Further, we extend the RST to a general framework (G-RST), where we can adjust regularization degree on sharpness freely for any scheduling function. We show that G-RST can outperform SAM in most cases while saving 50% extra computation cost.

## 1 INTRODUCTION

Deep neural networks (DNNs) have shown great capabilities in solving many real-world complex tasks (He et al., 2016; Redmon et al., 2016; Devlin et al., 2018). However, it is quite challenging to efficiently train them to achieve good performance, especially for today's severely overparameterized networks (Dosovitskiy et al., 2021; Han et al., 2017). Although such numerous parameters can improve the expressiveness of DNNs, yet they may complicate the geometry of the loss surface and generate more global and local minima in this huge hypothesis weight space.

By leveraging the finding that flat minima could exhibit better generalization ability, Foret et al. (2021) propose a sharpness-aware learning method called SAM, where loss geometry will be connected to the optimization to guide optimizers to converge to flat minima. Training with the SAM has shown the power to significantly improve model performance for various tasks (Foret et al., 2021; Chen et al., 2021). But on the other hand, the computation cost of SAM is almost *twice* that of the vanilla stochastic gradient descent (SGD), since it will incur one additional forward-backward propagation for each training iteration, which largely burdens the computation in practice.

Recently, techniques are introduced to improve the computation efficiency in SAM. Specifically, instead of using the full batch samples, Bahri et al. (2021) and Du et al. (2021a) select only part of batch samples to make approximations for the two forward-backward propagations. Although the computation cost can be reduced to some extent, unfortunately, the forward-backward propagation count in the SAM training scheme will not change essentially. Mi et al. (2022) randomly masking out part of weights during optimization in expectation to reduce the amount of gradient computations at each iteration. However, the efficiency improvement of such a method is strongly limited by the chain rule of gradient computation (Du et al., 2021a). Besides, Liu et al. (2022) propose to repeatedly use the past descent vertical gradients in SAM to reduce the incurred computational overhead.

Meanwhile, random selection strategy is a powerful technique for boosting optimization efficiency, particularly in the field of gradient boosting (Friedman, 2001), where a small set of learners in

gradient boosting machines would be selected randomly to be optimized under certain rule (Lu & Mazumder, 2020; Konstantinov et al., 2021).

Inspired by such randomization scheme in gradient boosting, we would present a simple but efficient training scheme, called *Randomized Sharpness-Aware Training* (RST). In our RST, the learning process would be randomized, where optimizers would randomly select to perform from base learning algorithms and sharpness-aware learning algorithms at each training iteration with a given probability. And this selecting probability is arranged by a custom scheduling function predefined before training. The scheduling function not only controls how much propagation count would be reduced, but also impacts the model performance.

Our contribution can be summarized as,

1. We propose a simple but efficient training scheme, called RST, which could reduce the propagation count via mixing base learning (SGD) algorithms and sharpness-aware learning (SAM) algorithms randomly.

2. We give interpretation of our RST scheme from the perspective of gradient norm regularization (GNR) (Zhao et al., 2022), and theoretically prove the convergence of RST scheme.

3. We empirically study the effect when arranging different scheduling functions, including totally three typical types of function families with six function groups.

4. We extend the RST to a general framework (G-RST), where GRN algorithm is mixed such that regularization degree on gradient norm can be adjusted freely. By training both CNN models and ViT models on commonly-used datasets, we show that G-RST can outperform SAM mostly while saving at least 50% extra computation cost.

## 1.1 OTHER RELATED WORKS

We would like to discuss works associated with the research on flat minima. In Hochreiter & Schmidhuber (1997), the authors are the first to point out that the flatness of minima could be associated with the model generalization, where models with better generalization should converge to flat minima. And such claim has been supported extensively by both empirical evidences and theoretical demonstrations (Keskar et al., 2017; Dinh et al., 2017). In the meantime, researchers are also fascinating by how to implement practical algorithms to force the models to converge to such flat minima. By summarizing this problem to a specific minimax optimization, Foret et al. (2021) introduce the SAM training scheme, which successfully guides optimizers to converge to flat minima. Further, Zheng et al. (2021) perform gradient descent twice to solve the minimization and maximization respectively in this minimax optimization. In Kwon et al. (2021), Adaptive SAM training scheme for improving SAM to be able to remain steady when performing weight rescaling operations. Zhao et al. (2022) seek flat minima by explicitly penalizing the gradient norm of the loss function. Unlike SAM-related training schemes, without a restriction on neighborhood region, Du et al. (2022) We propose to minimize the KL-divergence between the output distributions yielded by the current model and the moving average of past models, similar to the idea of knowledge distillation.

## 2 METHOD

### 2.1 RANDOMIZED SHARPNESS-AWARE TRAINING (RST)

The general idea of RST follows a randomization scheme, where the learning process will be randomized. Specifically, for each training iteration $t$, optimizers would perform a Bernoulli trial to choose from base learning algorithms and sharpness-aware learning algorithms. Here, we will consider first mixing the two most commonly-used algorithms, SGD and SAM. Thus, in each Bernoulli trial, the optimizer would perform the SAM algorithm with a probability $p(t)$ or perform the SGD algorithm with probability $1-p(t)$. Here, $p(t)$ could be a predefined custom function of iteration $t$, and we would call it the scheduling function of RST. Apparently, the sample space for this Bernoulli trial corresponds to the set $\Omega = \{\text{SGD, SAM}\}$. Correspondingly, a random variable could be defined on this sample space, $X(t) : \Omega \to \{0, 1\}$, where $X(t) = 0$ denotes performing the SGD algorithm while $X(t) = 1$ denotes performing the SAM algorithm. In summary, $X(t) \sim \text{Bernoulli}(p(t))$, and

$$\boldsymbol{\theta}_0 \xrightarrow{X(1)} \boldsymbol{\theta}_1 \cdots \boldsymbol{\theta}_t \xrightarrow{X(t+1)} \boldsymbol{\theta}_{t+1}, \ X(t) \in \{0, 1\} \tag{1}$$

---

**Algorithm 1** Randomized Sharpness-Aware Training (RST)

---

**Input**: Training set $\mathcal{S} = \{(\boldsymbol{x}_i, \boldsymbol{y}_i)\}_{i=0}^N$; loss function $L(\cdot)$; batch size $B$; learning rate $\alpha$; total iterations $T$; neighborhood radius of SAM $\rho$, scheduling function $p(t)$.
**Parameter**: Model weights $\boldsymbol{\theta}$.
**Output**: Optimized model weights $\hat{\boldsymbol{\theta}}$.
**Algorithm**:
  1: Initialize weight $\boldsymbol{\theta}_0$; initialize optimizer with scheduling function $p(t)$.
  2: **for** iteration $t = 1$ to $T$ **do**
  3:     Compute the gradient $\boldsymbol{g} = \nabla_{\boldsymbol{\theta}} L(\boldsymbol{\theta}_t)$.
  4:     Perform the Bernoulli trial with probability $p_t$ and record the result $X_t$.
  5:     **if** $X_t = 0$ **then**                                    ▷ *Implement SGD algorithm*
  6:         $\boldsymbol{g}_t = \boldsymbol{g}$.
  7:     **else**                                                     ▷ *Implement SAM algorithm*
  8:         $\boldsymbol{g}_t = \nabla_{\boldsymbol{\theta}} L(\boldsymbol{\theta}_t)$ at $\boldsymbol{\theta}_t = \boldsymbol{\theta}_t + \boldsymbol{\epsilon}_t$ with $\boldsymbol{\epsilon}_t = \rho \frac{g}{||g||}$.
  9:     **end if**
 10:     Update weight $\boldsymbol{\theta}_{t+1} = \boldsymbol{\theta}_t - \eta \cdot \boldsymbol{g}_t$
 11: **end for**
 12: **return** final weight $\hat{\boldsymbol{\theta}} = \boldsymbol{\theta}_T$

---

Additionally, Algorithm 1 shows the complete implementation when training with RST scheme.

Compared to the SAM training scheme, every time SGD algorithm is selected instead of SAM algorithm in the RST scheme, we would save one forward-backward propagation. Therefore, for training iteration $t$, the expectation of propagation count $\hat{\eta}_t$ in RST could be

$$\hat{\eta}_t = 2 \cdot p_t + 1 \cdot (1 - p_t) = 1 + p_t \tag{2}$$

Here, $p_t$ denotes the scheduling probability of $p(t)$ at training iteration $t$. Equation 2 indicates RST would incur extra more $p_t$ propagation count in expectation than the vanilla SGD training. Further, the average of the extra expected propagation count $\Delta\hat{\eta}$ over the total training iterations $T$ is,

$$\Delta\hat{\eta} = \frac{\sum_{t=0}^T (p_t)}{T} \tag{3}$$

where $\Delta\hat{\eta} \in [0, 1]$, bounded between $\Delta\hat{\eta}$ in the vanilla SGD scheme and the SAM scheme.

Obviously, the scheduling function $p(t)$ would straightforwardly control the number of propagations being saved. $\Delta\bar{\eta}$ would be larger if performing the SAM optimization with a higher probability. Also, an appropriate schedule could improve model performance further while a bad one may largely harm the training. We would provide a detailed study on the scheduling function in the later sections.

## 2.2 UNDERSTANDING RST FROM GRADIENT NORM REGULARIZATION

From previous demonstration, the gradient of RST at training iteration $t$ could be expressed as,

$$g_t = (1 - X_t) \cdot \nabla_{\boldsymbol{\theta}} L(\boldsymbol{\theta}_t) + X_t \nabla_{\boldsymbol{\theta}} L(\boldsymbol{\theta}_t + \boldsymbol{\epsilon}_t) \tag{4}$$

where $\boldsymbol{\epsilon}_t = \rho \cdot \nabla_{\boldsymbol{\theta}} L(\boldsymbol{\theta}_t) / ||\nabla_{\boldsymbol{\theta}} L(\boldsymbol{\theta}_t)||$. And the expectation of this gradient over $X$ is,

$$\mathbb{E}_X[g_t] = (1 - p_t) \nabla_{\boldsymbol{\theta}} L(\boldsymbol{\theta}_t) + p_t \nabla_{\boldsymbol{\theta}} L(\boldsymbol{\theta}_t + \boldsymbol{\epsilon}_t) \tag{5}$$

According to Zhao et al. (2022), gradients in the form of Equation 5 can be interpreted as regularizations on the gradient norm (GRN) of loss function.

Specifically, when imposing penalty on the gradient norm during training with a penalty coefficient $\gamma$, $L(\boldsymbol{\theta}) + \gamma ||\nabla_{\boldsymbol{\theta}} L(\boldsymbol{\theta})||$, the corresponding gradient could be approximated via the combination between $\nabla_{\boldsymbol{\theta}} L(\boldsymbol{\theta}_t)$ and $\nabla_{\boldsymbol{\theta}} L(\boldsymbol{\theta}_t + \boldsymbol{\epsilon}_t)$, which is

$$g_t^{(gnr)} = (1 - \frac{\gamma}{\rho}) \nabla_{\boldsymbol{\theta}} L(\boldsymbol{\theta}_t) + \frac{\gamma}{\rho} \nabla_{\boldsymbol{\theta}} L(\boldsymbol{\theta}_t + \boldsymbol{\epsilon}_t) \tag{6}$$

meaning that SAM is one special implementation of gradient norm regularization, where $\gamma_{\text{sam}} = \rho$.

From Equation 5 and Equation 6, we could reason that $p_t$ in Equation 5 has an equivalent effect with the term $\gamma/\rho$ in GNR. It means the equivalent penalty coefficient in RST would be

$$\gamma_{\text{rst}} = p_t \cdot \rho = p_t \cdot \gamma_{\text{sam}} \tag{7}$$

Compared to the SAM training scheme, the penalty degree is reduced by a factor of $p_t$ in RST.

## 2.3 CONVERGENCE ANALYSIS OF RST

In this section, we would give analysis in regards to the convergence in RST.

**Theorem 1.** *Assume the gradient of the loss function $L(\cdot)$ is $\beta$-smoothness, i.e. $||\nabla L(\boldsymbol{\theta}_1) - \nabla L(\boldsymbol{\theta}_2)|| \leq \beta||\boldsymbol{\theta}_1 - \boldsymbol{\theta}_2||$ for $\forall \boldsymbol{\theta}_1, \boldsymbol{\theta}_2 \in \Theta$. For iteration steps $T \geq 0$, learning rate $\alpha_t \leq 1/\beta$ and $\sqrt{p_t}\rho \leq 1/\beta$, we have*

$$\min_{t \in \{0,1,\cdots,T-1\}} ||\nabla L(\boldsymbol{\theta}_t)||^2 \leq \frac{2(L(\boldsymbol{\theta}_0) - L_*)}{\sum_{t \in \{0,1,\cdots,T-1\}} \alpha_t} + \frac{\sum_{t \in \{0,1,\cdots,T-1\}} \alpha_t p_t \rho^2 \beta^2}{\sum_{t \in \{0,1,\cdots,T-1\}} \alpha_t}$$

We would provide detailed proof in the Appendix. Basically, $||\nabla L(\boldsymbol{\theta})||^2 \leq \epsilon$ is generally used as one stopping criteria in optimization. The theorem implies that the minimum of $||\nabla L(\boldsymbol{\theta}_t)||^2$ over the training steps would reach such condition at a certain step within finite training steps.

**Corollary 1.** *For constant learning rate $\alpha_t = C/\beta$ or cosine learning rate schedules $\alpha_t = 2C/\beta \cdot (\frac{1}{2} + \frac{1}{2}\cos(\frac{t}{T}\pi))$, and constant scheduling probability $p_t = p$, we have*

$$\min_{t \in \{0,1,\cdots,T-1\}} ||\nabla L(\boldsymbol{\theta}_t)||^2 \leq \frac{2\beta(L(\boldsymbol{\theta}_0) - L_*)}{CT} + p\rho^2\beta^2$$

**Corollary 2.** *For decayed learning rate $\alpha_t = C/t$ and constant scheduling probability $p$, we have*

$$\min_{t \in \{0,1,\cdots,T-1\}} ||\nabla L(\boldsymbol{\theta}_t)||^2 \leq \frac{2(L(\boldsymbol{\theta}_0) - L_*)}{C \log T} + p\rho^2\beta^2$$

Corollary 1 and 2 show the convergence of common implementation in practice.

**Theorem 2.** *Assume the gradient of the loss function $L(\cdot)$ is $\beta$-smoothness. Assume Polyak-Lojasiewicz condition, i.e. $\frac{1}{2}||\nabla L(\boldsymbol{\theta}_t)||^2 \geq \varrho(L(\boldsymbol{\theta}_t) - L_*)$. For iteration steps $T \geq 0$, learning rate $\alpha_t \leq 1/\beta$ and $\sqrt{p_t}\rho \leq 1/\beta$, we have,*

$$\frac{\mathbb{E}_X[L(\boldsymbol{\theta}_t)] - L_*}{L(\boldsymbol{\theta}_0) - L_*} \leq \prod_{t \in \{0,1,\cdots,T-1\}} \left(1 - \alpha_t \varrho(1 - p_t \rho_t^2 \beta^2)\right)$$

Appendix shows the proof. Theorem 2 indicates that RST experiences a linear convergence rate.

## 3 EMPIRICAL STUDY OF SCHEDULING FUNCTION $p(t)$

In this section, we would investigate the computation efficiency and the impact on model performance when training with the RST scheme under different types of scheduling functions $p(t)$.

### 3.1 BASIC SETTING AND BASELINES

In our investigation of the effect of scheduling functions, we will train models with different scheduling functions from scratch to tackle the image classification tasks on Cifar-{10, 100} datasets, and compare the corresponding convergence performance and the incurred extra computation overhead.

For models, we would choose ResNet18 (He et al., 2016) and WideResNet-28-10 (Zagoruyko & Komodakis, 2016) architectures as our main target. For data augmentation, we would follow the basic strategy, where each image would be randomly flipped horizontally, then padded with four extra pixels and finally cropped randomly to $32 \times 32$. Expect for the scheduling functions implemented in the RST schemes, all the involved models are trained for 200 epochs with exactly the same hyperparameters. For each training case, we would run with five different seeds and report the average mean and standard deviation of these five runs. All the training details could be found in Appendix. Meanwhile, we have also reported additional results regarding other model architectures and other data augmentation strategy in Appendix.

Before our investigations on scheduling functions in RST, we would like to clarity the baseline first, where models are trained with the vanilla SGD scheme and SAM scheme. Table 1 shows the corresponding results, including the testing error rate (Error column), the training time (Time

Table 1: Testing error rate of ResNet18 and WideResNet28-10 models on Cifar10 and Cifar100 datasets when training with the SGD scheme and SAM scheme respectively.

| Model | Scheme | $\Delta\hat{\eta}_c$ | Cifar10 | | Cifar100 | |
| --- | --- | --- | --- | --- | --- | --- |
| | | | Time[m] | Error[%] | Time[m] | Error[%] |
| ResNet18 | SGD | $-$ | $15.8_{\pm0.4}$ | $4.48_{\pm0.10}$ | $15.4_{\pm0.3}$ | $20.79_{\pm0.12}$ |
| | SAM | 1.0 | $+16.0_{\pm0.5}$ | $3.81_{\pm0.07}$ | $+15.8_{\pm0.4}$ | $19.99_{\pm0.13}$ |
| WRN28-10 | SGD | $-$ | $33.3_{\pm0.5}$ | $3.53_{\pm0.10}$ | $33.7_{\pm0.6}$ | $18.69_{\pm0.12}$ |
| | SAM | 1.0 | $+27.4_{\pm0.3}$ | $2.78_{\pm0.07}$ | $+28.0_{\pm0.4}$ | $16.53_{\pm0.13}$ |

column) and the extra expected propagation count ($\Delta\hat{\eta}$ column). For the training time, we would report the total wall time spent to train for 200 epochs on four A100 Nvidia GPUs. From the table, we could find that compared to the SGD scheme, the SAM scheme could indeed improve the model performance, but in the meantime would incur more computations (102% for ResNet18 and 83% for WideResNet28-10).

## 3.2 IMPLEMENTATION OF SCHEDULING FUNCTION

Here, we will focus on studying three types of function families, which can cover most scheduling patterns. Table 2 shows the basic information regarding the three function scheduling families.

Table 2: Scheduling functions $p(t)$ and extra propagation counts $\Delta\hat{\eta}$ of the three function families.

| | Constant | Piecewise | Linear |
| --- | --- | --- | --- |
| Scheduling Function $p(t)$ | $a_c$ | $\begin{cases} a_p, & t \leq b_pT \\ 1-a_p, & t > b_pT \end{cases}$ | $a_lt + b_l$ |
| Propagation Count $\Delta\hat{\eta}$ | $a_c$ | $1 + 2a_pb_p - b_p - a_p$ | $p_l(\frac{T}{2})$ |

**Constant Function Family** In constant scheduling function family, the scheduling probability is $p_c(t) = a_c$, where $a_c \in [0, 1]$. Optimizers would select to perform the SAM algorithm with a fixed probability $a_c$ and the SGD algorithm with $1 - a_c$ during the whole training process. This implies that the extra computation overhead for constant scheduling function is proportional to the scheduling probability $a_c$.

Here, we will experimentally investigate a group of implementation with constant functions, where the scheduling probability $a_c$ will be set from 0.1 to 0.9 with an interval of 0.1. Figure 1A shows the scheduling functions of this group and Figure 1B shows the relationship between the extra expected propagation count $\hat{\eta}$ ($x$-axis) and the extra practical training wall time ($y$-axis) incurred by selecting SAM algorithm in RST. We could see that for both ResNet18 and WideResNet28-10 models, all the points locate very close to the reference line ($x = y$). The actual extra training wall time can be almost fully decided by the theoretical extra $\Delta\hat{\eta}$. Therefore, we could directly use $\Delta\hat{\eta}$ to indicate the extra computation cost for RST in the following demonstrations.

Then, Figure 1C shows the corresponding testing error rates of the two models with error bars (neighbor area) on Cifar10 (left) and Cifar100 (right). In the figure, $x$-axis denotes the extra $\Delta\hat{\eta}$ and meanwhile the markers are scaled by the actual training wall time. And the endpoints on both sides of the lines denote the testing error rates of training with the SGD scheme and the SAM scheme. Firstly, we could find that even with the lowest probability $a_c = 0.1$, as long as SAM algorithm is involved during training process, testing error rates could be generally reduced compared to those trained with only the SGD algorithm. But on the other side, model performance can not be improved continuously with the growth selecting probability towards the SAM algorithm. Secondly, compared to the SAM scheme, testing error rates would already reach comparable performance when $a_c = 0.6$ in RST, which would save about 40% computation overhead. In particular, when around $a_c = 0.8$, models would achieve the best performance, slightly outperforming the SAM scheme (3.65%/19.61% for ResNet18 and 2.71%/16.17% for WideResNet28-10 in RST). Additionally, we could see from the error bars that despite the randomness introduced in RST, training would still be fairly stable over the five runs.

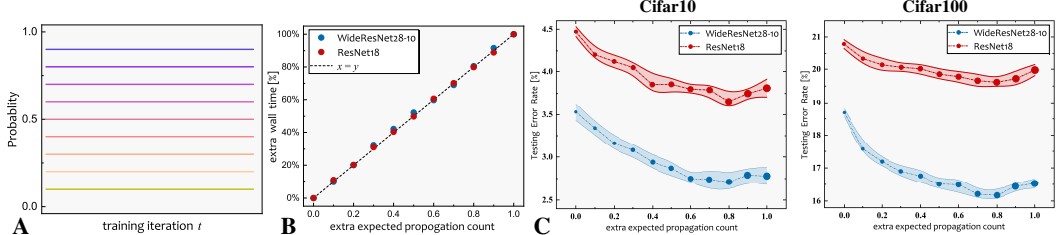

Figure 1: (A) Scheduling function plots for the constant scheduling function family. (B) Scatter plot between the extra expected propagation count $\hat{\eta}$ ($x$-axis) and the extra practical training wall time ($y$-axis) incurred in RST. (C) Testing error rates of ResNet18 and WideResNet-28-10 models with error bars on Cifar10 (left) and Cifar100 (right) when training with these constant scheduling functions. The markers are scaled by the training wall time.

**Piecewise Function Family**   Generally, the selecting probability in piecewise function would experience a stage conversion during training. In the first stage, optimizers would be arranged to perform SAM algorithm with a probability of $a_p$ in the beginning $b_pT$ training iterations, and then in the second stage, this probability would change to $1 - a_p$ for the rest training iterations. In our investigation, we would consider totally three typical groups of piecewise scheduling functions, where Figure 2 shows the corresponding scheduling function plots and their final results.

For the first group, we would set $a_p = 0$ and change the stage-related parameter $b_p$ from 0.1 to 0.9 with an interval of 0.1. Now, the optimizer actually behaves in a deterministic manner, which performs SGD algorithm in the first $b_pT$ iterations and then switches to SAM algorithm for the rest. Therefore, the larger $b_p$ is, the longer SGD algorithm will be performed, and the less extra computation overhead will be incurred. From the results, we could find that for all the training cases in this group, as implementing more iterations with SAM algorithm, we could get better performance gradually, which could achieve better performance than those trained with the SAM scheme. And the best performance between this group and the constant group are very close (3.66%/19.47% for ResNet18 and 2.69%/16.31% for WideResNet28-10 in this group).

Next, in the second group, we would arrange training in an opposite way from piecewise group 1, where we will keep all the settings except deploying $a_p = 1$. Optimizers would perform SAM algorithm in the first $b_pT$ iterations and then switch to SGD for the rest steps. Actually, models could not get good performance under such arrangement. The results show that training needs to accumulate sufficient SAM iterations to completely outperform SGD scheme. Models could reach competitive performance only when performing SGD algorithm in the last few iterations. Intuitively, implementation pattern of piecewise group 2 would somewhat go against the core of sharpness-aware learning. Frequently implementing SGD algorithm near the end of training would be harmful to the convergence to flat minima.

Unlike previous patterns, in piecewise group 3, we fix $b_p = 0.5$ and change $a_p$ from 0.1 to 0.9 with an interval of 0.1. Now, optimizers will pick SAM algorithm with probability $a_p$ for the first half of training iterations and then switch the probability to $1 - a_p$ for the rest. For all the training instances in this group, we have $\Delta\hat{\eta} = 0.5$. And, the actual training wall time between these cases are rather close (Time[m]: +8.2($\pm$0.4) for ResNet18 and +13.9($\pm$0.7) for WideResNet28-10). Note that the results of this group are plotted against the evolution of $a_p$, not the propagation count. We could see in the results that model performance would gradually get higher as the growth probability of implementation with SAM algorithm in the second stage. This somehow again confirms the previous demonstration of avoiding frequently implementing SGD algorithm near the end of training.

**Linear Function Family**   For linear scheduling functions, the selecting probability $p(t)$ is scheduled linearly, changing monotonously with either an increasing or a decreasing pattern. Optimizers would select to perform SAM algorithm with decreasing probability when $a_l \le 0$ while with increasing probability when $a_l \ge 0$. Notably, from the summary table 2, the computation overhead of such implementation is actually decided by the scheduling probability at $T/2$. We would focus on two typical groups of linear scheduling functions in our experiments. Figure 2 show the scheduling functions and the results.

In the first group, we would schedule the functions to pass through two given points, where the first point is $(T/2, m)$ and the second point is either $(0, 0)$ or $(1, 1)$ depending on the value of $m$. Here, the parameter $m$ denotes the probability to be set at the training iteration $T/2$. And we would

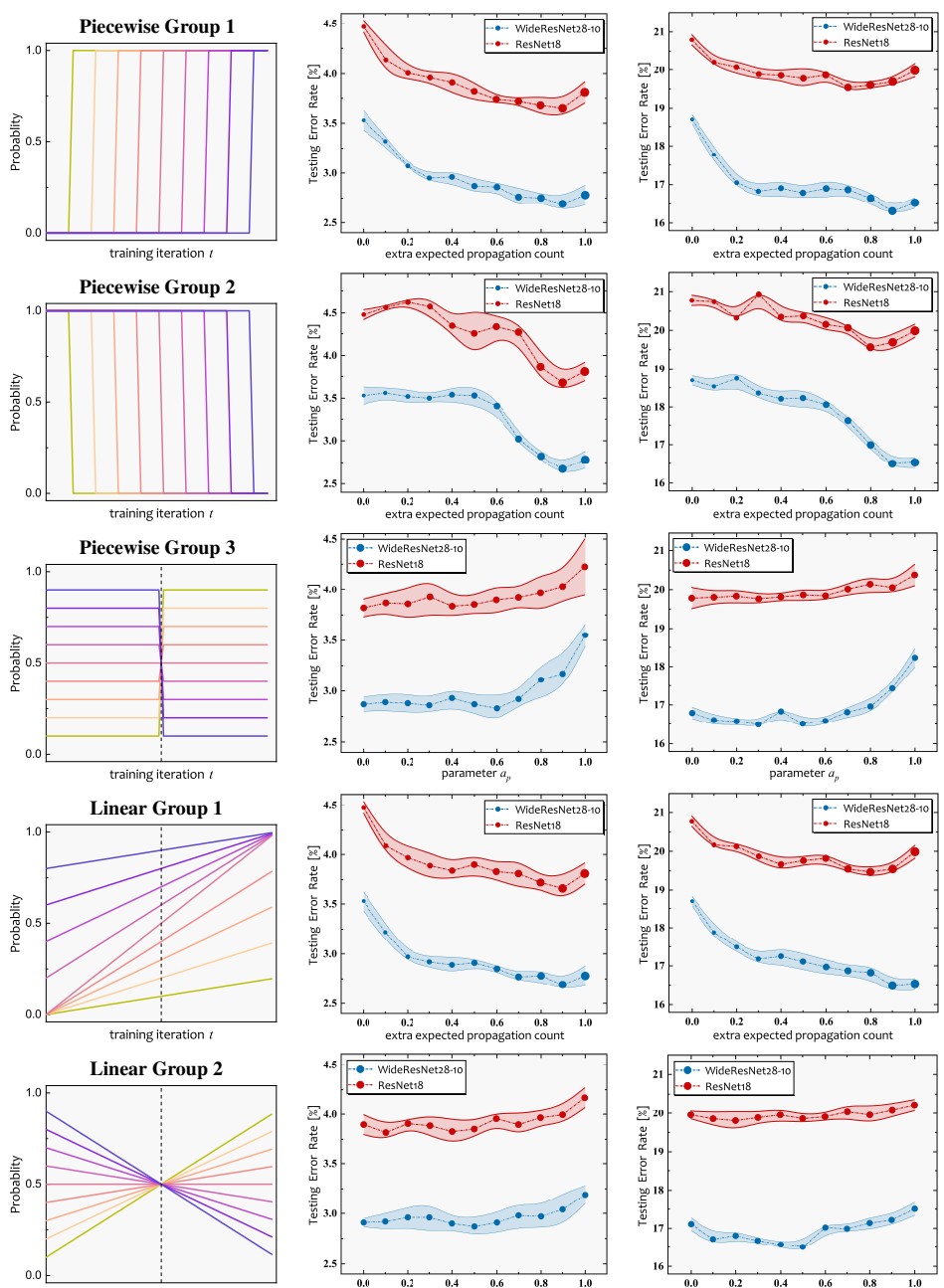

Figure 2: Testing error rates of ResNet18 and WideResNet-28-10 models with error bars on Cifar10 (left) and Cifar100 (right) when training separately with the three groups of piecewise scheduling functions and the two groups of linear scheduling functions in RST.

set it from 0.1 to 0.9 with an interval of 0.1. Clearly, in this group, the probability of selecting SAM algorithm would increase over the iterations. Also, as $m$ increases, SAM algorithm would experience an overall higher probability of selection. We could find in the results that as performing more SAM algorithm, model performance would be more and more better. And the trend of model performance in this group would be quite similar to that in piecewise group 1. Actually, these two groups share very close selection patterns in general, where the scheduling probability is changed instantaneously in piecewise group 1 while it becomes gradually in this group.

As for the second group, the scheduling functions would pass through two points that are $(T/2, 0.5)$ and $(0, b_l)$. This means that training will always incur 0.5 extra propagation count in expectation, $\Delta \overline{\eta} = 0.5$. From the results, we can find that similar to those in piecewise group 3, model perfor-

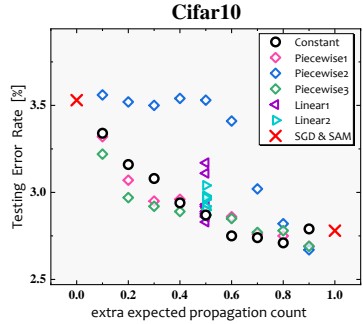 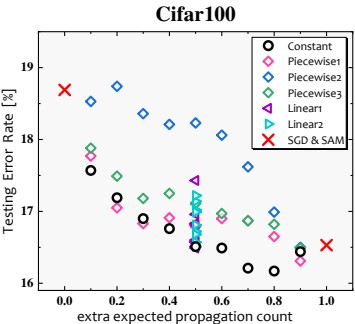

Figure 3: Summary plot of model performance in regards to the extra computation overhead for all the scheduling function cases.

mance will also progressively become higher, but more mildly. Likewise, the two groups also have close selection pattern, as in the same way of that between piecewise group 1 and linear group 1.

### 3.3 SUMMARY

To give a summary view of these scheduling functions, Figure 3 gives the scatter plot of WideResNet28-10 between the model performance and the incurred extra propagation counts for all the scheduling function cases. From previous demonstrations and the figure, we can conclude,

- Avoid to schedule the SGD algorithm with relatively high probability near the end of training since it would largely harm the training.
- Generally, scheduling SAM algorithm with higher probability in total would bring better model performance, where the best model performance in RST would outperform those in SAM scheme.
- Compared to other schedules, simple constant scheduling functions could give decent model performance. So, we recommend using constant scheduling functions in practice for both their simplicity and effectiveness.

## 4 GENERAL FRAMEWORK FOR RST

Recall that from Equation 7, SAM training is actually regularizing the gradient norm with $\gamma_{\text{sam}} = \rho$, and RST to mix SGD algorithm and SAM algorithm would have a scaling effect on this penalty by a factor of $p_t$. However, when the scheduling probability $p_t$ is low, RST may be unable to provide sufficient equivalent regularization effect on gradient norm. This motivates to expand RST to a general form (G-RST) which mixes between SGD algorithm and GNR algorithm (Equation 6) such that G-RST could freely adjust the scaling effect of the penalty degree on gradient norm,

$$g_t = (1 - X_t) \cdot g_t^{(sgd)} + X_t \cdot g_t^{(gnr)} \tag{8}$$

In this way, G-RST would be given an extra freedom to control the scaled penalty degree via $\gamma$ in GNR, which would be $\gamma_{\text{rst}} = p_t \gamma_{\text{gnr}}$. It allows training to impose arbitrary regularization on gradient norm while enjoying a high probability of selecting SGD algorithm.

In our following experiments, we would use the constant scheduling functions because of their efficiency and simplicity as demonstrated previously. Here, we would consider $p(t) = 0.5$, so we need to set $\gamma_{\text{gnr}} = 2$ when mixing, to provide an equivalent regularization as that in SAM scheme.

We would first train models with G-RST on Cifar datasets, which involves both CNN models and ViT models (Dosovitskiy et al., 2021). For CNN models, we would keep the basic settings the same as those in the previous section. As for ViT models, we would train each case for 1200 epochs and adopt some further data augmentation to get the best performance. Note that the base algorithm switch to Adam in ViT models. All the training details are reported in the Appendix.

Table 3 shows the corresponding results of these models on Cifar datasets. We could observe from the table that compared to the SAM scheme, G-RST could improve the model performance further to some extent while saving 50% of the extra computation overhead for all the training cases. This indicates that adjusting the penalty coefficient in RST can give comparable effect as that in SAM.

Table 3: Testing error rate of CNN models and ViT models on Cifar10 and Cifar100 datasets when training with SGD, SAM and the G-RST where $p(t) = 0.5$.

| | Learning Methods | C-10&100 Time[m] | Cifar10 Error[%] | Cifar100 Error[%] |
|---|---|---|---|---|
| VGG16BN | SGD | $9.9_{\pm 0.2}$ | $5.74_{\pm 0.09}$ | $25.22_{\pm 0.31}$ |
| | SAM | $+8.9_{\pm 0.3}$ | $5.24_{\pm 0.08}$ | $24.23_{\pm 0.29}$ |
| | G-RST[50%] | $+4.4_{\pm 0.5}$ | $5.21_{\pm 0.08}$ | $24.37_{\pm 0.29}$ |
| ResNet18 | SGD | $15.6_{\pm 0.3}$ | $4.48_{\pm 0.10}$ | $20.79_{\pm 0.12}$ |
| | SAM | $+15.9_{\pm 0.4}$ | $3.81_{\pm 0.07}$ | $19.99_{\pm 0.13}$ |
| | G-RST[50%] | $+7.9_{\pm 0.3}$ | $3.65_{\pm 0.10}$ | $19.95_{\pm 0.18}$ |
| WRN28-10 | SGD | $33.5_{\pm 0.5}$ | $3.53_{\pm 0.10}$ | $18.99_{\pm 0.12}$ |
| | SAM | $+27.7_{\pm 0.4}$ | $2.78_{\pm 0.07}$ | $16.53_{\pm 0.13}$ |
| | G-RST[50%] | $+14.3_{\pm 0.6}$ | $2.68_{\pm 0.05}$ | $16.19_{\pm 0.15}$ |
| Pyramid164 | SGD | $119.7_{\pm 1.1}$ | $3.42_{\pm 0.09}$ | $17.82_{\pm 0.15}$ |
| | SAM | $+83.2_{\pm 0.9}$ | $2.61_{\pm 0.07}$ | $14.80_{\pm 0.18}$ |
| | G-RST[50%] | $+42.0_{\pm 1.2}$ | $2.50_{\pm 0.11}$ | $14.55_{\pm 0.21}$ |
| ViT-Ti16 | Adam | $189.0_{\pm 1.8}$ | $9.45_{\pm 0.18}$ | $34.79_{\pm 0.27}$ |
| | SAM | $+165.2_{\pm 2.4}$ | $8.59_{\pm 0.16}$ | $32.48_{\pm 0.31}$ |
| | G-RST[50%] | $+82.9_{\pm 2.5}$ | $8.31_{\pm 0.18}$ | $32.17_{\pm 0.24}$ |
| ViT-S16 | Adam | $247.9_{\pm 2.9}$ | $6.89_{\pm 0.17}$ | $27.48_{\pm 0.32}$ |
| | SAM | $+263.1_{\pm 2.1}$ | $5.52_{\pm 0.20}$ | $26.53_{\pm 0.27}$ |
| | G-RST[50%] | $+131.9_{\pm 3.3}$ | $5.39_{\pm 0.14}$ | $26.24_{\pm 0.28}$ |
| ViT-B16 | Adam | $407.8_{\pm 2.9}$ | $6.56_{\pm 0.23}$ | $27.95_{\pm 0.28}$ |
| | SAM | $+400.2_{\pm 2.1}$ | $5.45_{\pm 0.17}$ | $26.51_{\pm 0.30}$ |
| | G-RST[50%] | $+199.6_{\pm 3.3}$ | $5.58_{\pm 0.20}$ | $26.27_{\pm 0.26}$ |

Table 4: Testing error rate of CNN models on ImageNet datasets when training with SGD, SAM and the G-RST where $p(t) = 0.5$.

| | Methods | Time[m] | Top-1[%] | Top-5[%] |
|---|---|---|---|---|
| ResNet50 | SGD | $750_{\pm 9}$ | $23.64_{\pm 0.17}$ | $7.01_{\pm 0.09}$ |
| | SAM | $+518_{\pm 5}$ | $23.16_{\pm 0.11}$ | $6.72_{\pm 0.06}$ |
| | G-RST[50%] | $+259_{\pm 12}$ | $22.82_{\pm 0.19}$ | $6.63_{\pm 0.11}$ |
| ResNet101 | SGD | $1255_{\pm 11}$ | $21.93_{\pm 0.09}$ | $6.11_{\pm 0.07}$ |
| | SAM | $+904_{\pm 8}$ | $21.02_{\pm 0.10}$ | $5.31_{\pm 0.09}$ |
| | G-RST[50%] | $+451_{\pm 14}$ | $20.78_{\pm 0.12}$ | $5.16_{\pm 0.10}$ |

Following the same setting of $p_t$ as that on Cifar datasets, we would train ResNet-{50, 101} models on ImageNet for 100 epochs to further investigate the effectiveness of G-RST on large-scale dataset. Table 4 shows the final results, where each case is trained over three random seeds. Likewise, we can find that G-RST can also give better model performance while being 50% less computational expensive than SAM scheme, which again confirms the effectiveness of G-RST.

## 5 CONCLUSION

We propose a simple but efficient training scheme, called Randomized Sharpness-Aware Training (RST), for reducing the computation overhead in the sharpness-aware training. In RST, optimizers will be scheduled to randomly select from the base learning algorithm and sharpness-aware learning training scheme at each training iteration. Such a scheme can be interpreted as regularization on gradient norm with scaling effect. Then, we theoretically prove RST converges in finite training iterations. As for the scheduling functions, we empirically show that simple constant scheduling functions can achieve comparable results with other scheduling functions. Finally, we extend the RST to a general framework (G-RST), where the regularization effect can be adjusted freely. We show that G-RST can outperform SAM to some extent while reducing 50% extra computation cost.

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

# A  PROOF OF THEOREM 1 & 2

## A.1  PROOF OF THEOREM 1

In randomized sharpness-aware training (RST), weights $\boldsymbol{\theta}_t$ are updated stochastically with a random variable $X_t \sim B(1, p_t)$,

$$\boldsymbol{\theta}_{t+1} = \boldsymbol{\theta}_t - \alpha_t \nabla L(\boldsymbol{\theta}_t + X_t \rho_t \nabla L(\boldsymbol{\theta}_t)) \tag{9}$$

For $\beta$-smoothness functions, we have

$$L(\boldsymbol{\theta}_1) \leq L(\boldsymbol{\theta}_2) + \nabla L(\boldsymbol{\theta}_2)^T (\boldsymbol{\theta}_1 - \boldsymbol{\theta}_2) + \frac{\beta}{2} ||\boldsymbol{\theta}_1 - \boldsymbol{\theta}_2||^2 \tag{10}$$

Then, we set $\boldsymbol{\theta}_1 = \boldsymbol{\theta}_{t+1}$ and $\boldsymbol{\theta}_2 = \boldsymbol{\theta}_t$,

$$L(\boldsymbol{\theta}_{t+1}) \leq L(\boldsymbol{\theta}_t) + \langle \nabla L(\boldsymbol{\theta}_t), \boldsymbol{\theta}_{t+1} - \boldsymbol{\theta}_t \rangle + \frac{\beta}{2} ||\boldsymbol{\theta}_{t+1} - \boldsymbol{\theta}_t||^2$$

$$\leq L(\boldsymbol{\theta}_t) - \langle \nabla L(\boldsymbol{\theta}_t), \alpha_t \nabla L(\boldsymbol{\theta}_t + X_t \rho_t \nabla L(\boldsymbol{\theta}_t)) \rangle + \frac{\beta}{2} ||\alpha_t \nabla L(\boldsymbol{\theta}_t + X_t \rho_t \nabla L(\boldsymbol{\theta}_t))||^2$$

$$\leq L(\boldsymbol{\theta}_t) - \alpha_t \langle \nabla L(\boldsymbol{\theta}_t), \nabla L(\boldsymbol{\theta}_t + X_t \rho_t \nabla L(\boldsymbol{\theta}_t)) \rangle + \frac{\alpha_t^2 \beta}{2} ||\nabla L(\boldsymbol{\theta}_t + X_t \rho_t \nabla L(\boldsymbol{\theta}_t))||^2 \tag{11}$$

For $\alpha_t \leq 1/\beta$,

$$L(\boldsymbol{\theta}_{t+1}) \leq L(\boldsymbol{\theta}_t) - \alpha_t \langle \nabla L(\boldsymbol{\theta}_t), \nabla L(\boldsymbol{\theta}_t + X_t \rho_t \nabla L(\boldsymbol{\theta}_t)) \rangle + \frac{\alpha_t}{2} ||\nabla L(\boldsymbol{\theta}_t + X_t \rho_t \nabla L(\boldsymbol{\theta}_t))||^2 \tag{12}$$

Next, add $\frac{\alpha_t}{2}||\nabla L(\boldsymbol{\theta}_t)||^2$ and subtract $\frac{\alpha_t}{2}||\nabla L(\boldsymbol{\theta}_t)||^2$,

$$
\begin{aligned}
L(\boldsymbol{\theta}_{t+1}) &\leq L(\boldsymbol{\theta}_t) + \frac{\alpha_t}{2}||\nabla L(\boldsymbol{\theta}_t)||^2 - \alpha_t \langle \nabla L(\boldsymbol{\theta}_t), \nabla L(\boldsymbol{\theta}_t + X_t \rho_t \nabla L(\boldsymbol{\theta}_t)) \rangle \\
&\quad + \frac{\alpha_t}{2}||\nabla L(\boldsymbol{\theta}_t + X_t \rho_t \nabla L(\boldsymbol{\theta}_t))||^2 - \frac{\alpha_t}{2}||\nabla L(\boldsymbol{\theta}_t)||^2 \\
&\leq L(\boldsymbol{\theta}_t) + \frac{\alpha_t}{2}||\nabla L(\boldsymbol{\theta}_t + X_t \rho_t \nabla L(\boldsymbol{\theta}_t)) - \nabla L(\boldsymbol{\theta}_t)||^2 - \frac{\alpha_t}{2}||\nabla L(\boldsymbol{\theta}_t)||^2 \\
&\leq L(\boldsymbol{\theta}_t) + \frac{\alpha_t}{2}||\beta X_t \rho_t \nabla L(\boldsymbol{\theta}_t)||^2 - \frac{\alpha_t}{2}||\nabla L(\boldsymbol{\theta}_t)||^2 \\
&\leq L(\boldsymbol{\theta}_t) - \frac{\alpha_t}{2}(1 - X_t^2 \rho_t^2 \beta^2)||\nabla L(\boldsymbol{\theta}_t)||^2
\end{aligned}
\tag{13}
$$

So, $\rho_t \leq 1/\beta$ such that the loss would decrease continuously in training,

$$
L(\boldsymbol{\theta}_{t+1}) \leq L(\boldsymbol{\theta}_t) \leq L(\boldsymbol{\theta}_{t-1}) \cdots \leq L(\boldsymbol{\theta}_0)
\tag{14}
$$

Rearrange Equation 13,

$$
\frac{\alpha_t}{2}(1 - X_t^2 \rho_t^2 \beta^2)||\nabla L(\boldsymbol{\theta}_t)||^2 \leq L(\boldsymbol{\theta}_t) - L(\boldsymbol{\theta}_{t+1})
\tag{15}
$$

Taking expectation gives,

$$
\begin{aligned}
\mathbb{E}_X \left[ \frac{\alpha_t}{2}(1 - X_t^2 \rho_t^2 \beta^2)||\nabla L(\boldsymbol{\theta}_t)||^2 \right] &\leq \mathbb{E}_X[L(\boldsymbol{\theta}_t)] - \mathbb{E}_X[L(\boldsymbol{\theta}_{t+1})] \\
(1 - p_t)\frac{\alpha_t}{2}||\nabla L(\boldsymbol{\theta}_t)||^2 + p_t \frac{\alpha_t}{2}(1 - \rho_t^2 \beta^2)||\nabla L(\boldsymbol{\theta}_t)||^2 &\leq \mathbb{E}_X[L(\boldsymbol{\theta}_t)] - \mathbb{E}_X[L(\boldsymbol{\theta}_{t+1})] \\
\frac{\alpha_t}{2}(1 - p_t \rho_t^2 \beta^2)||\nabla L(\boldsymbol{\theta}_t)||^2 &\leq \mathbb{E}_X[L(\boldsymbol{\theta}_t)] - \mathbb{E}_X[L(\boldsymbol{\theta}_{t+1})]
\end{aligned}
\tag{16}
$$

For $\rho_t = \rho/||\nabla L(\boldsymbol{\theta}_t)||$ in SAM optimization, the Equation 16,

$$
\begin{aligned}
\frac{\alpha_t}{2}(1 - p_t \rho_t^2 \beta^2)||\nabla L(\boldsymbol{\theta}_t)||^2 &\leq \mathbb{E}_X[L(\boldsymbol{\theta}_t)] - \mathbb{E}_X[L(\boldsymbol{\theta}_{t+1})] \\
\frac{\alpha_t}{2}||\nabla L(\boldsymbol{\theta}_t)||^2 - \frac{\alpha_t p_t \rho_t^2 \beta^2}{2}||\nabla L(\boldsymbol{\theta}_t)||^2 &\leq \mathbb{E}_X[L(\boldsymbol{\theta}_t)] - \mathbb{E}_X[L(\boldsymbol{\theta}_{t+1})] \\
\frac{\alpha_t}{2}||\nabla L(\boldsymbol{\theta}_t)||^2 - \frac{\alpha_t p_t \rho^2 \beta^2}{2||\nabla L(\boldsymbol{\theta}_t)||^2}||\nabla L(\boldsymbol{\theta}_t)||^2 &\leq \mathbb{E}_X[L(\boldsymbol{\theta}_t)] - \mathbb{E}_X[L(\boldsymbol{\theta}_{t+1})] \\
\frac{\alpha_t}{2}||\nabla L(\boldsymbol{\theta}_t)||^2 &\leq \mathbb{E}_X[L(\boldsymbol{\theta}_t)] - \mathbb{E}_X[L(\boldsymbol{\theta}_{t+1})] + \frac{\alpha_t p_t \rho^2 \beta^2}{2}
\end{aligned}
\tag{17}
$$

Then, sum over the training steps,

$$
\sum_{t \in \{0,1,\cdots,T-1\}} \frac{\alpha_t}{2}||\nabla L(\boldsymbol{\theta}_t)||^2 \leq L(\boldsymbol{\theta}_0) - L_* + \sum_{t \in \{0,1,\cdots,T-1\}} \frac{\alpha_t p_t \rho^2 \beta^2}{2}
\tag{18}
$$

Here, $L(\boldsymbol{\theta}_0)$ is the loss of the initialization model and $L_*$ denotes the optimal point, $L_* = \min L(\boldsymbol{\theta})$.
Since $\min_{t \in \{0,1,\cdots,T-1\}} ||\nabla L(\boldsymbol{\theta}_t)||^2 \leq ||L(\boldsymbol{\theta})||^2$, we have,

$$
\min_{t \in \{0,1,\cdots,T-1\}} ||\nabla L(\boldsymbol{\theta}_t)||^2 \leq \frac{2(L(\boldsymbol{\theta}_0) - L_*)}{\sum_{t \in \{0,1,\cdots,T-1\}} \alpha_t} + \Xi
\tag{19}
$$

where,

$$
\Xi = \frac{\sum_{t \in \{0,1,\cdots,T-1\}} \alpha_t p_t \rho^2 \beta^2}{\sum_{t \in \{0,1,\cdots,T-1\}} \alpha_t}
\tag{20}
$$

Generally, Equation 19 indicates that for $\epsilon$-suboptimal termination criteria $||L(\boldsymbol{\theta}_t)|| \leq \epsilon$, hybrid training would satisfy such convergence condition in finite training steps.

Further, for constant learning rate schedules $\alpha_t = C/\beta$ or cosine learning rate schedules $\alpha_t = 2C/\beta \cdot (\frac{1}{2} + \frac{1}{2} \cos(\frac{t}{T}\pi))$, and constant scheduling functions $p_t = p$, we have

$$\min_{t \in \{0,1,\cdots,T-1\}} ||\nabla L(\boldsymbol{\theta}_t)||^2 \leq \frac{2\beta(L(\boldsymbol{\theta}_0) - L_*)}{CT} + p\rho^2\beta^2 \tag{21}$$

Here, we use $\sum_{t=0}^{T} \cos(\frac{t}{T}\pi) = 0$, which we would prove in the following lemma. In other words, the epsilon $\epsilon$ is associated with the $\mathcal{O}(1/T)$.

For decayed learning rate schedule $\alpha_t = C/t$, and constant scheduling functions $p_t = p$, we have

$$\min_{t \in \{0,1,\cdots,T-1\}} ||\nabla L(\boldsymbol{\theta}_t)||^2 \leq \frac{2(L(\boldsymbol{\theta}_0) - L_*)}{C \log T} + p\rho^2\beta^2 \tag{22}$$

In other words, the epsilon $\epsilon$ is associated with the $\mathcal{O}(1/\log T)$.

**Lemma 1.** *For $t \in \{0, 1, 2, \cdots, T\}$, we have*

$$\sum_{t=0}^{T} \cos(\frac{t}{T}\pi) = 0 \tag{23}$$

**Proof** For trigonometric functions,

$$\sum_{t=0}^{T} g(\frac{t}{T}\pi) \tag{24}$$

where $g \in \{\sin, \cos\}$. We would use the Euler's identity,

$$e^{ix} = \cos x + i \sin x \tag{25}$$

Therefore, we have $\cos x = \Re\{e^{ix}\}$ and $\sin x = \Im\{e^{ix}\}$, where $\Re\{\cdot\}$ and $\Im\{\cdot\}$ denote the real part and imaginary part.

In this way, for $g = \cos$, Equation 24 would be,

$$\begin{aligned}
\sum_{t=0}^{T} \cos(\frac{t}{T}\pi) &= \sum_{t=0}^{T} \Re\{e^{i\frac{t}{T}\pi}\} \\
&= \Re\{\sum_{t=0}^{T} e^{i\frac{t}{T}\pi}\} \\
&= \Re\{\frac{e^0(1 - e^{i\frac{T+1}{T}\pi})}{1 - e^{i\frac{1}{T}\pi}}\} \\
&= \Re\{\frac{e^{i\frac{T+1}{2T}\pi} \cdot (e^{-i\frac{T+1}{2T}\pi} - e^{i\frac{T+1}{2T}\pi})}{e^{i\frac{1}{2T}\pi} \cdot (e^{-i\frac{1}{2T}\pi} - e^{i\frac{1}{2T}\pi})}\} \\
&= \Re\{e^{i\frac{T}{2T}\pi} \frac{\sin(\frac{T+1}{2T}\pi)}{\sin(\frac{1}{2T})\pi}\} \\
&= \cos(\frac{T}{2T}\pi)\frac{\sin(\frac{T+1}{2T}\pi)}{\sin(\frac{1}{2T})\pi} \\
&= 0 \quad (\cos\frac{\pi}{2} = 0)
\end{aligned} \tag{26}$$

$\square$

## A.2 PROOF OF THEOREM 2

From the Polyak-Lojasiewicz condition,

$$\frac{1}{2}||\nabla L(\boldsymbol{\theta}_t)||^2 \geq \varrho(L(\boldsymbol{\theta}_t) - L_*) \tag{27}$$

From the previous Equation 16, we would have,

$$\frac{\alpha_t}{2}(1 - p_t\rho_t^2\beta^2)||\nabla L(\boldsymbol{\theta}_t)||^2 \leq \mathbb{E}_X\left[L(\boldsymbol{\theta}_t)\right] - \mathbb{E}_X\left[L(\boldsymbol{\theta}_{t+1})\right]$$

$$\alpha_t\varrho(1 - p_t\rho_t^2\beta^2)(\mathbb{E}_X\left[L(\boldsymbol{\theta}_t)\right] - L_*) \leq \mathbb{E}_X\left[L(\boldsymbol{\theta}_t)\right] - \mathbb{E}_X\left[L(\boldsymbol{\theta}_{t+1})\right]$$

$$\alpha_t\varrho(1 - p_t\rho_t^2\beta^2)(\mathbb{E}_X\left[L(\boldsymbol{\theta}_t)\right] - L_*) \leq (\mathbb{E}_X\left[L(\boldsymbol{\theta}_t)\right] - L_*) - (\mathbb{E}_X\left[L(\boldsymbol{\theta}_{t+1})\right] - L_*) \tag{28}$$

$$\frac{\mathbb{E}_X\left[L(\boldsymbol{\theta}_{t+1})\right] - L_*}{\mathbb{E}_X\left[L(\boldsymbol{\theta}_t)\right] - L_*} \leq 1 - \alpha_t\varrho(1 - p_t\rho_t^2\beta^2)$$

Then, performing iterative multiplication over the training steps gives,

$$\frac{\mathbb{E}_X\left[L(\boldsymbol{\theta}_t)\right] - L_*}{L(\boldsymbol{\theta}_0) - L_*} \leq \prod_{t\in\{0,1,\cdots,T-1\}}\left(1 - \alpha_t\varrho(1 - p_t\rho_t^2\beta^2)\right) \tag{29}$$

End of the proof. $\square$

## B  ADDITIONAL RESULTS

### B.1  TRIGONOMETRIC SCHEDULING FUNCTION

We would like to use WideResNet28-10 to further investigate the scheduling functions which are trigonometric functions $p_{tr}(t)$ in RST. Here, we would confine the trigonometric functions to only sinusoidal functions and cosine functions. And more specifically, we focus on investigating four scheduling functions,

$$\begin{cases} p_{cos1}(t) &= \frac{1}{2} + \frac{1}{2}\cos\frac{t}{T}\pi \\ p_{cos2}(t) &= 1 - p_{cos1}(t) = \frac{1}{2} - \frac{1}{2}\cos\frac{t}{T}\pi \\ p_{sin1}(t) &= \sin\frac{t}{T}\pi \\ p_{sin2}(t) &= 1 - p_{sin1}(t) = 1 - \sin\frac{t}{T}\pi \end{cases} \tag{30}$$

Note that all these functions are in the range between 0 and 1.

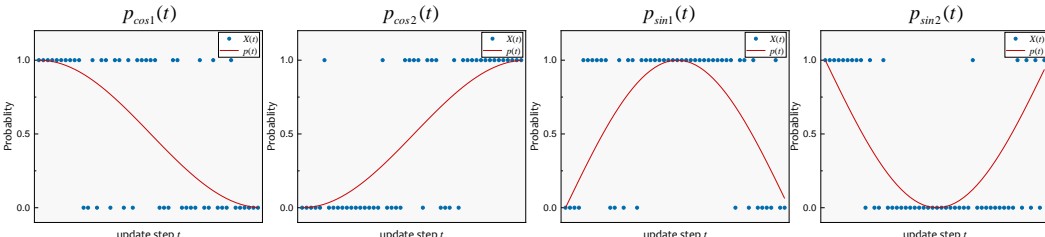

Figure 4: Scheduling function plots for the four trigonometric scheduling functions. The blue points stand for the instance of random variable $X$.

Table 5: Testing error rate of WideResNet28-10 models on Cifar10 and Cifar100 datasets when training with the four trigonometric scheduling functions.

| WideResNet28-10 | Cifar-10&100 | | Cifar10 | Cifar100 |
|---|---|---|---|---|
| | Time[m] | $\Delta\hat{\eta}$ | Error[%] | Error[%] |
| SGD | $33.5_{\pm0.5}$ | $0$ | $3.53_{\pm0.10}$ | $18.99_{\pm0.12}$ |
| SAM | $+27.7_{\pm0.4}$ | $1$ | $2.78_{\pm0.07}$ | $16.53_{\pm0.13}$ |
| RST: $p_{cos1}(t)$ | $+14.7_{\pm0.6}$ | $0.5$ | $3.16_{\pm0.09}$ | $17.08_{\pm0.12}$ |
| RST: $p_{cos2}(t)$ | $+14.9_{\pm0.7}$ | $0.5$ | $2.86_{\pm0.10}$ | $16.77_{\pm0.18}$ |
| RST: $p_{sin1}(t)$ | $+17.6_{\pm0.4}$ | $2/\pi \approx 0.63$ | $2.81_{\pm0.09}$ | $16.69_{\pm0.12}$ |
| RST: $p_{sin2}(t)$ | $+17.9_{\pm0.6}$ | $2/\pi$ | $3.21_{\pm0.13}$ | $17.15_{\pm0.10}$ |

Figure 4 shows the training scheme plots of the four functions and Table 5 shows the final results. From the table, when training with these trigonometric scheduling functions, training will incur 50% extra expected average propagation count for cosine functions and $\pi/2 \approx 64\%$ for sinusoidal functions.

For cosine functions, we could find that their pattern of scheduling probability could be quite close to linear functions. This could lead to that they may yield very similar performances. As for sinusoidal functions, implementations would present monotonously increasing or decreasing probability for the first half iterations and then switch to the opposite for the rest. Compared to that of cosine functions, as SAM would be implemented with more frequency in total, the corresponding results would be better. Additionally, the results have also confirmed that the performance would be degenerate when SGD is frequently selected near the end of training. And in summary, training with such complex trigonometric scheduling functions could not present better results than that with simple constant scheduling functions. We would still recommend to use simple constant scheduling functions in practical implementation.

### B.2 $\gamma_{rst}$ IN G-RST

Based on the demonstrations on the G-RST, we would know that G-RST could adjust the regularization effect on the gradient norm freely for a given selecting probability. Therefore, we would perform some more tuning on the $\gamma_{\text{gnr}}$ to be mixed in RST to present the relationship between the model performance and the equivalent regularization degree $\gamma_{\text{rst}}$. Here we would perform a grid searching over the selecting probability from 0.1 to 0.9 with an interval of 0.2, and then set the $\gamma_{\text{gnr}}$ (Equation 9 in the main paper) in the RST to fix the equivalent regularization effect $\gamma_{\text{rst}}$ across 0.5 to 1.5.

Table 5 shows the corresponding 2D plot. From the table, we could find that when the selecting probability $p_t$ is very low, even if we impose a high regularization penalty, models could not be trained to achieve good performance. This is mainly because that based on $\gamma_{\text{rst}} = \gamma_{\text{gnr}} p_t$, for these low $p_t$, we have to mix a very high $\gamma_{\text{gnr}}$ to get a fair equivalent effect $\gamma_{\text{rst}}$. When the $\gamma_{\text{gnr}}$ is very high in GNR, according to the paper Zhao et al. (2022), it would cause a lose of precision on the approximations on the Hessian multiplication. Secondly, we could also find from the figure that when the equivalent regularization degree $\gamma_{\text{rst}}$ is around the range from 0.8 to 1, models could achieve the better performances than others. Imposing too much regularization on the gradient norm would instead harm the performance. For the fixed $\gamma_{\text{rst}}$, increasing the selecting probability $p_t$ would somewhat improve the model performance, but not in a significant manner.

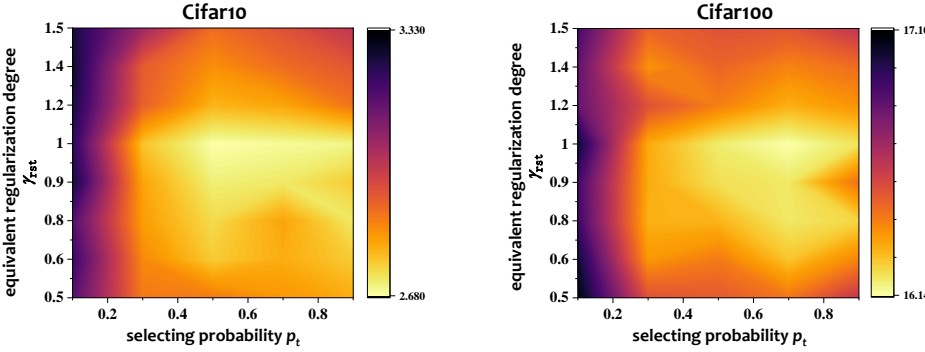

Figure 5: 2D image plot between the selecting probability $p_t$ and the equivalent regularization degree $\gamma_{\text{rst}}$ for WideResNet28-10 when training with RST.

In summary, it is recommended to set a moderate selecting probability and combine with a proper $\gamma_{\text{gnr}}$ that could lead to $\gamma_{\text{rst}}$ around. In this way, training would enjoy a gain on the computation efficiency and give satisfactory performance at the same time. And the Table 3 in the main paper actually follows

## B.3 EXPERIMENT RESULTS WHEN USING CUTOUT REGULARIZATION

In addition to the basic data augmentation strategy used in the previous section, we would also investigate the effect when using the Cutout Regularization Devries & Taylor (2017). Here, we would choose WideResNet28-10 as our main experiment target. Also, the training hyperparameters are the same as them used in the previous sections.

The tables below show the final results, where trainings are going to be separately scheduled by constant scheduling functions (Table 6), the first group of piecewise scheduling functions (Table 7) and the first group of linear scheduling functions (Table 8) and trigonometric scheduling functions (Table 9). From the results, we would come to the same conclusions as those in the summary sections. In short, constant scheduling functions would be a good choice for practical implementation, which would be simple to implement and be able to yield at least comparable performance to other scheduling functions.

Table 6: Testing error rate of WideResNet28-10 models on Cifar10 and Cifar100 datasets with Cutout regularization when training with constant scheduling functions.

| Training Scheme | Cifar-10 & 100 $\Delta\hat{\eta}_c$ | Cifar10 Error[%] | Cifar100 Error[%] |
|---|---|---|---|
| SGD | $-$ | $2.81_{\pm 0.07}$ | $16.91_{\pm 0.10}$ |
| SAM | $1.0$ | $2.43_{\pm 0.13}$ | $14.87_{\pm 0.16}$ |
| RST | $0.1$ | $2.67_{\pm 0.05}$ | $16.14_{\pm 0.18}$ |
| | $0.2$ | $2.53_{\pm 0.06}$ | $15.97_{\pm 0.16}$ |
| | $0.3$ | $2.46_{\pm 0.06}$ | $15.56_{\pm 0.14}$ |
| | $0.4$ | $2.40_{\pm 0.06}$ | $15.17_{\pm 0.22}$ |
| | $0.5$ | $2.32_{\pm 0.06}$ | $15.10_{\pm 0.11}$ |
| | $0.6$ | $2.31_{\pm 0.07}$ | $14.96_{\pm 0.08}$ |
| | $0.7$ | $2.25_{\pm 0.08}$ | $14.94_{\pm 0.09}$ |
| | $0.8$ | $2.23_{\pm 0.03}$ | $14.81_{\pm 0.09}$ |
| | $0.9$ | $2.31_{\pm 0.06}$ | $14.71_{\pm 0.03}$ |

Table 7: Testing error rate of WideResNet28-10 models on Cifar10 and Cifar100 datasets with Cutout regularization when training with constant scheduling functions.

| Training Scheme | Cifar-10 & 100 $\Delta\hat{\eta}_c$ | Cifar10 Error[%] | Cifar100 Error[%] |
|---|---|---|---|
| SGD | $-$ | $2.81_{\pm 0.07}$ | $16.91_{\pm 0.10}$ |
| SAM | $1.0$ | $2.43_{\pm 0.13}$ | $14.87_{\pm 0.16}$ |
| RST | $0.1$ | $2.69_{\pm 0.04}$ | $15.70_{\pm 0.21}$ |
| | $0.2$ | $2.51_{\pm 0.07}$ | $15.37_{\pm 0.18}$ |
| | $0.3$ | $2.47_{\pm 0.05}$ | $15.38_{\pm 0.33}$ |
| | $0.4$ | $2.44_{\pm 0.04}$ | $15.24_{\pm 0.35}$ |
| | $0.5$ | $2.46_{\pm 0.02}$ | $15.24_{\pm 0.22}$ |
| | $0.6$ | $2.34_{\pm 0.02}$ | $14.98_{\pm 0.28}$ |
| | $0.7$ | $2.34_{\pm 0.03}$ | $14.99_{\pm 0.26}$ |
| | $0.8$ | $2.30_{\pm 0.05}$ | $14.84_{\pm 0.09}$ |
| | $0.9$ | $2.29_{\pm 0.05}$ | $14.75_{\pm 0.24}$ |

Table 8: Testing error rate of WideResNet28-10 models on Cifar10 and Cifar100 datasets with Cutout regularization when training with constant scheduling functions.

| Training Scheme | Cifar-10 & 100 $\Delta\hat{\eta}_c$ | Cifar10 Error[%] | Cifar100 Error[%] |
|---|---|---|---|
| SGD | − | $2.81_{\pm 0.07}$ | $16.91_{\pm 0.10}$ |
| SAM | 1.0 | $2.43_{\pm 0.13}$ | $14.87_{\pm 0.16}$ |
| | 0.1 | $2.63_{\pm 0.02}$ | $15.83_{\pm 0.27}$ |
| | 0.2 | $2.58_{\pm 0.06}$ | $15.67_{\pm 0.16}$ |
| | 0.3 | $2.45_{\pm 0.05}$ | $15.33_{\pm 0.17}$ |
| | 0.4 | $2.37_{\pm 0.07}$ | $15.30_{\pm 0.19}$ |
| RST | 0.5 | $2.38_{\pm 0.02}$ | $15.33_{\pm 0.11}$ |
| | 0.6 | $2.27_{\pm 0.10}$ | $14.97_{\pm 0.02}$ |
| | 0.7 | $2.25_{\pm 0.03}$ | $14.95_{\pm 0.23}$ |
| | 0.8 | $2.22_{\pm 0.03}$ | $14.68_{\pm 0.14}$ |
| | 0.9 | $2.23_{\pm 0.10}$ | $14.79_{\pm 0.04}$ |

Table 9: Testing error rate of WideResNet28-10 models on Cifar10 and Cifar100 datasets with Cutout regularization when training with constant scheduling functions.

| Training Scheme | Cifar-10 & 100 $\Delta\hat{\eta}_c$ | Cifar10 Error[%] | Cifar100 Error[%] |
|---|---|---|---|
| SGD | − | $2.81_{\pm 0.07}$ | $16.91_{\pm 0.10}$ |
| SAM | 1.0 | $2.43_{\pm 0.13}$ | $14.87_{\pm 0.16}$ |
| $p_{cos2}(t)$ | 0.5 | $2.35_{\pm 0.03}$ | $15.02_{\pm 0.19}$ |
| $p_{sin1}(t)$ | 0.5 | $2.27_{\pm 0.04}$ | $14.85_{\pm 0.17}$ |

## B.4 ADDITIONAL EXPERIMENT RESULTS FOR OTHER MODELS

Other than ResNet18 and WideResNet28-10, we would also investigate another model architecture, including the VGG16 Simonyan & Zisserman (2015) with batch normalization and Vision Transformer. From the previous results, we could see that the constant scheduling functions would already provide representative results. So here we would only investigate the results when trained with constant scheduling functions to make comparisons with the baselines.

Table 10: Testing error rate of VGG16-BN models on Cifar10 and Cifar100 datasets when training with constant scheduling functions.

| Training Scheme | Cifar-10 & 100 $\Delta\hat{\eta}_c$ | Cifar10 Error[%] | Cifar100 Error[%] |
|---|---|---|---|
| SGD | − | $5.74_{\pm 0.09}$ | $25.22_{\pm 0.31}$ |
| SAM | 1.0 | $5.24_{\pm 0.08}$ | $24.23_{\pm 0.29}$ |
| | 0.1 | $5.59_{\pm 0.08}$ | $25.10_{\pm 0.21}$ |
| | 0.2 | $5.45_{\pm 0.07}$ | $24.97_{\pm 0.17}$ |
| | 0.3 | $5.39_{\pm 0.07}$ | $24.88_{\pm 0.12}$ |
| | 0.4 | $5.38_{\pm 0.03}$ | $24.73_{\pm 0.20}$ |
| RST | 0.5 | $5.35_{\pm 0.05}$ | $24.50_{\pm 0.17}$ |
| | 0.6 | $5.30_{\pm 0.07}$ | $24.42_{\pm 0.11}$ |
| | 0.7 | $5.29_{\pm 0.04}$ | $24.31_{\pm 0.18}$ |
| | 0.8 | $5.14_{\pm 0.05}$ | $24.17_{\pm 0.14}$ |
| | 0.9 | $5.21_{\pm 0.06}$ | $24.08_{\pm 0.15}$ |

Table 11: Testing error rate of ViT-S16 models on Cifar10 and Cifar100 datasets when training with constant scheduling functions. Note that the hyperparameters is different from those in the previous section. Here we only train for 300 epochs without mixup augmentation.

| Training Scheme | Cifar-10 & 100 $\Delta\hat{\eta}_c$ | Cifar10 Error[%] | Cifar100 Error[%] |
|---|---|---|---|
| SGD | $-$ | $12.59_{\pm 0.54}$ | $37.82_{\pm 0.31}$ |
| SAM | 1.0 | $11.91_{\pm 0.59}$ | $36.40_{\pm 0.26}$ |
| RST | 0.1 | $11.94_{\pm 0.55}$ | $37.21_{\pm 0.25}$ |
| | 0.2 | $11.79_{\pm 0.47}$ | $37.10_{\pm 0.29}$ |
| | 0.3 | $11.40_{\pm 0.54}$ | $36.64_{\pm 0.28}$ |
| | 0.4 | $11.17_{\pm 0.53}$ | $36.58_{\pm 0.17}$ |
| | 0.5 | $11.37_{\pm 0.24}$ | $36.52_{\pm 0.20}$ |
| | 0.6 | $11.31_{\pm 0.50}$ | $36.10_{\pm 0.17}$ |
| | 0.7 | $10.85_{\pm 0.11}$ | $36.36_{\pm 0.07}$ |
| | 0.8 | $11.78_{\pm 0.79}$ | $36.20_{\pm 0.20}$ |
| | 0.9 | $11.78_{\pm 0.67}$ | $36.38_{\pm 0.15}$ |

We could see in the table that RST again could boost the computational efficiency and in the meantime acquire better model generalization compared to that trained using the SAM scheme.

### B.5 USING RST SCHEME ON OTHER SAM VARIANTS

In this section, we are going to further show the effectiveness of our RST on SAM variants, where we would use ASAM (Kwon et al., 2021) and GSAM (Zhuang et al., 2022) as our investigation target. For both ASAM and GSAM, we would compare them with using our RST and G-RST schemes. Here, based on the previous demonstrations, the selecting probability in RST and G-RST is set constantly to $0.5$. And for G-RST, since the essence of these SAM variants is regularizing the gradient norm, we would double the regularization effect in G-RST, the same as the implementations in previous experiments. Table 12 shows the final results.

As we could see in the table, when using RST on ASAM and GSAM, we could obtain a similar results as using RST on SAM. Specifically, since RST and G-RST randomly selecting between sharpness-aware learning algorithm and the base learning algorithm, the computational efficiency could be largely improved for both ASAM and GSAM. And as previous demonstrations, RST would weaken the regularization effect, so we could see that the corresponding performance would be relatively lower than the standard sharpness-aware training. When doubling the regularization effect in G-RST, we could get comparable results with the standard sharpness-aware training, which again confirms the effectiveness of our method.

### B.6 MIXING RST SCHEME WITH OTHER EFFICIENT SAM TECHNIQUES

In RST, the optimizer would choose to perform the base learning algorithm and the sharpness-aware algorithm. When selecting sharpness-aware algorithm, we could meanwhile adopt other efficient techniques to further improve the training efficiency. Here, we would study the mixing effect of RST with separately LookSAM (Liu et al., 2022) and weight masking techniques (Mi et al., 2022; Du et al., 2021b). Table 13 shows the corresponding results.

As we could see in the table, for all these efficient techniques, our RST could improve the computational efficiency further. However, if the selecting probability in RST is relatively low (0.5 in the table), it may harm the mixing effect. On the other hand, as properly raising the selecting probability (0.75 in the table), it is possible to acquire comparable results with these efficient techniques.

Table 12: Testing error rate of CNN models and ViT models on Cifar10 and Cifar100 datasets when training with SGD, SAM and the G-RST where $p(t) = 0.5$.

| | Learning Methods | C-10&100 Time[m] | Cifar10 Error[%] | Cifar100 Error[%] |
|---|---|---|---|---|
| VGG16BN | ASAM | $+9.0_{\pm 0.2}$ | $5.28_{\pm 0.06}$ | $24.08_{\pm 0.14}$ |
| | ASAM & RST | $+4.4_{\pm 0.2}$ | $5.50_{\pm 0.10}$ | $24.49_{\pm 0.16}$ |
| | ASAM & G-RST | $+4.5_{\pm 0.3}$ | $5.32_{\pm 0.09}$ | $24.11_{\pm 0.25}$ |
| | GSAM | $+9.8_{\pm 0.4}$ | $5.74_{\pm 0.09}$ | $25.22_{\pm 0.31}$ |
| | GSAM & RST | $+4.7_{\pm 0.3}$ | $5.24_{\pm 0.08}$ | $24.23_{\pm 0.29}$ |
| | GSAM & G-RST | $+4.8_{\pm 0.4}$ | $5.21_{\pm 0.08}$ | $24.37_{\pm 0.29}$ |
| ResNet18 | ASAM | $+15.8_{\pm 0.3}$ | $3.77_{\pm 0.05}$ | $20.02_{\pm 0.15}$ |
| | ASAM & RST | $+8.0_{\pm 0.2}$ | $3.91_{\pm 0.09}$ | $20.31_{\pm 0.11}$ |
| | ASAM & G-RST | $+7.9_{\pm 0.3}$ | $3.65_{\pm 0.10}$ | $19.95_{\pm 0.18}$ |
| | GSAM | $+17.4_{\pm 0.3}$ | $3.81_{\pm 0.04}$ | $19.91_{\pm 0.13}$ |
| | GSAM & RST | $+8.9_{\pm 0.4}$ | $3.99_{\pm 0.03}$ | $20.43_{\pm 0.17}$ |
| | GSAM & G-RST | $+8.9_{\pm 0.3}$ | $3.70_{\pm 0.11}$ | $20.10_{\pm 0.18}$ |
| WRN28-10 | ASAM | $+27.6_{\pm 0.3}$ | $3.53_{\pm 0.10}$ | $16.40_{\pm 0.16}$ |
| | ASAM & RST | $+14.4_{\pm 0.3}$ | $2.78_{\pm 0.07}$ | $16.81_{\pm 0.15}$ |
| | ASAM & G-RST | $+14.3_{\pm 0.5}$ | $2.68_{\pm 0.05}$ | $16.59_{\pm 0.19}$ |
| | GSAM | $+30.6_{\pm 0.6}$ | $2.74_{\pm 0.04}$ | $16.51_{\pm 0.08}$ |
| | GSAM & RST | $+15.5_{\pm 0.5}$ | $2.95_{\pm 0.04}$ | $16.95_{\pm 0.15}$ |
| | GSAM & G-RST | $+15.7_{\pm 0.4}$ | $2.71_{\pm 0.07}$ | $16.47_{\pm 0.12}$ |

Table 13: Testing error rate of CNN models and ViT models on Cifar10 and Cifar100 datasets when training with SGD, SAM and the G-RST where $p(t) = 0.5$.

| | Learning Methods | C-10&100 Time[m] | Cifar10 Error[%] | Cifar100 Error[%] |
|---|---|---|---|---|
| **VGG16BN** | SGD | $9.9_{\pm 0.2}$ | $5.74_{\pm 0.09}$ | $25.22_{\pm 0.31}$ |
| | LookSAM(5)[1] | $+3.0_{\pm 0.2}$ | $5.49_{\pm 0.06}$ | $24.71_{\pm 0.26}$ |
| | LookSAM(5) & G-RST[50%] | $+1.6_{\pm 0.3}$ | $5.56_{\pm 0.09}$ | $24.88_{\pm 0.19}$ |
| | LookSAM(5) & G-RST[75%] | $+2.4_{\pm 0.3}$ | $5.30_{\pm 0.11}$ | $24.34_{\pm 0.23}$ |
| | SGD | $16.7_{\pm 0.5}$ | $6.12_{\pm 0.09}$ | $25.56_{\pm 0.22}$ |
| | ESAM[2] | $+13.5_{\pm 0.6}$ | $5.50_{\pm 0.07}$ | $24.49_{\pm 0.21}$ |
| | ESAM & G-RST[50%] | $+6.9_{\pm 0.4}$ | $5.92_{\pm 0.06}$ | $24.91_{\pm 0.16}$ |
| | ESAM & G-RST[70%] | $+10.2_{\pm 0.5}$ | $5.38_{\pm 0.12}$ | $24.57_{\pm 0.18}$ |
| | SGD | $16.7_{\pm 0.5}$ | $6.12_{\pm 0.09}$ | $25.56_{\pm 0.22}$ |
| | SSAM[2] | $+15.5_{\pm 0.4}$ | $5.64_{\pm 0.09}$ | $24.61_{\pm 0.17}$ |
| | SSAM & G-RST[50%] | $+8.1_{\pm 0.5}$ | $5.99_{\pm 0.12}$ | $25.03_{\pm 0.25}$ |
| | SSAM & G-RST[75%] | $+12.0_{\pm 0.6}$ | $5.59_{\pm 0.07}$ | $24.66_{\pm 0.21}$ |
| **ResNet18** | SGD | $15.6_{\pm 0.3}$ | $4.48_{\pm 0.10}$ | $20.79_{\pm 0.12}$ |
| | LookSAM(5) | $+5.6_{\pm 0.4}$ | $4.06_{\pm 0.09}$ | $20.30_{\pm 0.16}$ |
| | LookSAM(5) & G-RST[50%] | $+3.0_{\pm 0.2}$ | $4.18_{\pm 0.08}$ | $20.44_{\pm 0.27}$ |
| | LookSAM(5) & G-RST[75%] | $+4.4_{\pm 0.3}$ | $3.94_{\pm 0.12}$ | $20.11_{\pm 0.23}$ |
| | SGD | $24.4_{\pm 0.6}$ | $4.66_{\pm 0.05}$ | $20.98_{\pm 0.20}$ |
| | ESAM | $+18.6_{\pm 0.4}$ | $4.05_{\pm 0.07}$ | $20.28_{\pm 0.14}$ |
| | ESAM & G-RST[50%] | $+9.5_{\pm 0.6}$ | $4.41_{\pm 0.04}$ | $20.72_{\pm 0.12}$ |
| | ESAM & G-RST[75%] | $+14.0_{\pm 0.5}$ | $4.08_{\pm 0.08}$ | $20.21_{\pm 0.23}$ |
| | SGD | $24.4_{\pm 0.6}$ | $4.66_{\pm 0.05}$ | $20.98_{\pm 0.20}$ |
| | SSAM | $+21.0_{\pm 0.4}$ | $3.89_{\pm 0.04}$ | $20.17_{\pm 0.17}$ |
| | SSAM & G-RST[50%] | $+10.7_{\pm 0.3}$ | $4.03_{\pm 0.07}$ | $20.41_{\pm 0.13}$ |
| | SSAM & G-RST[75%] | $+15.8_{\pm 0.8}$ | $3.83_{\pm 0.09}$ | $20.19_{\pm 0.22}$ |
| **WRN28-10** | SGD | $33.5_{\pm 0.5}$ | $3.53_{\pm 0.10}$ | $18.99_{\pm 0.12}$ |
| | LookSAM(5) | $+9.4_{\pm 0.6}$ | $3.15_{\pm 0.11}$ | $17.47_{\pm 0.25}$ |
| | LookSAM(5) & G-RST[50%] | $+4.9_{\pm 0.4}$ | $3.22_{\pm 0.08}$ | $17.55_{\pm 0.18}$ |
| | LookSAM(5) & G-RST[75%] | $+7.1_{\pm 0.5}$ | $3.04_{\pm 0.10}$ | $17.09_{\pm 0.26}$ |
| | SGD | $109.9_{\pm 1.2}$ | $3.97_{\pm 0.05}$ | $19.13_{\pm 0.18}$ |
| | ESAM | $+91.4_{\pm 0.9}$ | $2.96_{\pm 0.06}$ | $16.90_{\pm 0.31}$ |
| | ESAM & G-RST[50%] | $+45.9_{\pm 1.4}$ | $3.20_{\pm 0.09}$ | $17.58_{\pm 0.20}$ |
| | ESAM & G-RST[75%] | $+69.2_{\pm 1.8}$ | $2.99_{\pm 0.10}$ | $16.94_{\pm 0.22}$ |
| | SGD | $109.9_{\pm 1.2}$ | $3.97_{\pm 0.05}$ | $19.13_{\pm 0.18}$ |
| | SSAM | $+107.3_{\pm 1.7}$ | $3.10_{\pm 0.04}$ | $16.97_{\pm 0.15}$ |
| | SSAM & G-RST[50%] | $+58.3_{\pm 2.1}$ | $3.24_{\pm 0.06}$ | $17.11_{\pm 0.24}$ |
| | SSAM & G-RST[75%] | $+81.2_{\pm 1.9}$ | $3.12_{\pm 0.10}$ | $16.56_{\pm 0.20}$ |

[1] Following the paper (Liu et al., 2022), LookSAM(5) denotes that update the descent gradient in SAM algorithm every five implementation iterations.

[2] Unlike LookSAM, ESAM and SSAM are both implemented on the git repository https://github.com/Mi-Peng/Sparse-Sharpness-Aware-Minimization, where one A100 GPU is used. And SGD baseline is also obtained based on this repository.

## C    TRAINING DETAILS

Code is available at github.com/JustNobody0204/Submission-ICLR2023. The basic training hyper-parameters are deployed as below,

Table 14: The basic hyperparameters for training CNNs on Cifar dataset.

|                            | SGD Scheme | SAM Scheme | RST Scheme |
| -------------------------- | ---------- | ---------- | ---------- |
| Epoch                      | 200        | 200        | 200        |
| Batch size                 | 256        | 256        | 256        |
| Base optimizer type        | SGD        | SGD        | SGD        |
| Basic learning rate        | 0.1        | 0.1        | 0.1        |
| Learning rate schedule     | cosine     | cosine     | cosine     |
| Weight decay               | 0.001      | 0.001      | 0.001      |
| Weight decay (PyramidNet)  | 0.0005     | 0.0005     | 0.0005     |
| $\rho$ in SAM              | -          | 0.1        | 0.1        |

Table 15: The basic hyperparameters for training ViTs.

|                         | Adam Scheme | SAM Scheme | RST Scheme |
| ----------------------- | ----------- | ---------- | ---------- |
| Data augmentation       | mixup       | mixup      | mixup      |
| Epoch                   | 1200        | 1200       | 1200       |
| Warmup epoch            | 40          | 40         | 40         |
| Batch size              | 256         | 256        | 256        |
| Base optimizer type     | Adam        | Adam       | Adam       |
| Basic learning rate     | 0.0005      | 0.0005     | 0.0005     |
| Learning rate schedule  | cosine      | cosine     | cosine     |
| Weight decay            | 0.03        | 0.03       | 0.03       |
| $\rho$ in SAM           | -           | 0.1        | 0.1        |

Table 16: The basic hyperparameters for training CNNs on ImageNet dataset.

|                         | SGD Scheme | SAM Scheme | RST Scheme |
| ----------------------- | ---------- | ---------- | ---------- |
| Epoch                   | 100        | 100        | 100        |
| Batch size              | 512        | 512        | 512        |
| Base optimizer type     | SGD        | SGD        | SGD        |
| Basic learning rate     | 0.2        | 0.2        | 0.2        |
| Learning rate schedule  | cosine     | cosine     | cosine     |
| Weight decay            | 0.0001     | 0.0001     | 0.0001     |
| $\rho$ in SAM           | -          | 0.05       | 0.05       |

