# OpenReview forum: "Randomized Sharpness-Aware Training for Boosting Computational Efficiency in Deep Learning"
_ICLR.cc/2023/Conference — Submitted to ICLR 2023_

### Official Review · Reviewer_5xxY · 2022-10-23

**Confidence:** 4
**Correctness:** 3
**Technical Novelty And Significance:** 2
**Empirical Novelty And Significance:** 2
**Recommendation:** 5

**Clarity, Quality, Novelty And Reproducibility:**

The paper is well written. The proposed method is clearly presented. However, the proposed method is not novel enough and the experiments are not solid enough.

**Strength And Weaknesses:**

Strength:
1. The motivation of the proposed method is clear. SAM is indeed suffering from low computational efficiency.
2. The proposed method is clearly presented. The reproducibility is not an issue.

Weakness:
1. The proposed algorithm is intuitive and not novel enough. It is adding a probability to replace some of the SAM steps with normal SGD optimization steps. This would not be an issue if the proposed method is verified to be super-effective. However,
2. The experiments do not show the superiority of the proposed method. The paper has missed some relevant baselines as mentioned in the summary section. Also, I was confused that why the selected architectures are different on CIFAR and ImageNet dataset?

**Summary Of The Paper:**

The paper targets at improving the efficiency of sharpness aware optimizer (SAM) by assigning a probability where a normal SGD optimizer is used. The decision is based on Bernoulli trial. The authors further propose a general framework to make the proposed optimizer schedule usable for different architectures. The idea is intuitive and clearly expressed. The proposed method is simple. This would not be an issue if it is extremely effective. However, the paper has missed many efficient SAM baselines such as GSAM [1] and SAF [2]. Besides, the experiments on large dataset ImageNet is not solid. The selected baseline models are different from the experiments on CIFAR-10 and the are not representative enough.


[1] Juntang Zhuang, Boqing Gong, Liangzhe Yuan, Yin Cui, Hartwig Adam, Nicha Dvornek, Sekhar Tatikonda, James Duncan, and Ting Liu. Surrogate gap minimization improves sharpness-aware training.

[2] Du, Jiawei, et al. "Sharpness-Aware Training for Free." arXiv preprint arXiv:2205.14083 (2022).

**Summary Of The Review:**

In summary, the proposed method is not well supported by the presented experiments mainly due to two points:

1. The missing baselines on efficient SAM algorithms and
2. The missing architectures on ImageNet dataset.

---

> ### Author Response · Authors · 2022-11-12
> **We thank the reviewer for the valuable and constructive comments! (Part 1)**
>
> `Q: The proposed algorithm is intuitive and not novel enough. It is adding a probability to replace some of the SAM steps with normal SGD optimization steps. This would not be an issue if the proposed method is verified to be super-effective.`
>
> We would like to thank the reviewer for the helpful comment. We understand the reviewer's concern. In deep learning, we generally want the method to be as simple as possible because we could easy to implement in practice. And on the other side, we should also give such simple method a comprehensive analysis to show how and why it works. The reviewer could find that we have provided a comprehensive demonstration about the meaning of the proposed RSL and theoretically proved the convergence of such randomized policy, not just in an intuitive manner. Based on the interpretation from the perspective of gradient norm regularization, it is possible for us to manually control the randomized effect in RST. Further, we have also presented a detailed empirical study of how should one schedule the probability in practical training. The presented results have shown the effectiveness of our methods.
>
> `Q: However, the paper has missed many efficient SAM baselines such as GSAM [1] and SAF [2]..`
>
> We would like to thank the reviewer for the valuable comment. Firstly, GSAM, ICLR2022. GSAM is an equivalent variant of SAM, which targets to contribute to the the convergence of SAM not its computational efficiency. Compared to SAM, GSAM still requires two steps i.e. two forward-backward propagations at each training iterations. For the two steps, GSAM keep the ascent step unchanged, and additionally subtract the orthogonal direction of gradient as the decent gradient.
>
> Since GSAM is not targeting to reduce the computational overhead in SAM, it is therefore not necessary to compare with GSAM. Instead, we could actually apply our randomized policy on these SAM variants like GSAM to boost their computational efficiency. The following table only shows the results of ResNet18. And for detailed results, the reviewer could check the appendix B.5 in the newly uploaded version, including the results regarding VGG16BN and WideResNet28-10.
>
>
> |Method|Time|C-10 Error|C-100 Error|
> |--|:--:|:--:|:--:|
> |SGD|15.6|4.48|20.79|
> |ASAM|+15.8|3.77|20.02|
> |ASAM + RST[50%]|+8.0|3.91|20.31|
> |ASAM + G-RST[50%]|+7.9|3.65|19.95|
> |GSAM|+17.4|3.81|19.91|
> |GSAM + RST[50%]|+8.9|3.99|20.43|
> |GSAM + G-RST[50%]|+8.9|3.70|20.10|
>
>
> As we could see in the table, when using RST on ASAM and GSAM, we could obtain a similar results as using RST on SAM. Specifically, since RST and G-RST randomly selecting between sharpness-aware learning algorithm and the base learning algorithm, the computational efficiency could be largely improved for both ASAM and GSAM. And based on our demonstrations, RST would weaken the regularization effect, so we could see that the corresponding performance would be relatively lower than the standard sharpness-aware training. When doubling the regularization effect in G-RST, we could get comparable results with the standard sharpness-aware training, which confirms the effectiveness of our method.

---

> > ### Author Response · Authors · 2022-11-12
> > **We thank the reviewer for the valuable and constructive comments! (Part 2)**
> >
> >
> > Secondly, "Sharpness-Aware Training for Free" (SAF), Nips 2022. SAF proposes to minimize the KL-divergence between the output distributions yielded by the current model and the moving average of past models. It should be particularly noted that unlike SAM, SAF follows another way, where no explicit regularization is imposed within the neighborhood region during the whole training process. In fact, this makes SAF seem more like the idea of knowledge distillation than sharpness-aware learning, where the teacher model is the moving average of past models and the student model is the current model. Besides, just as the results reported in their paper, the superiority of SAF over SAM variants still requires further demonstrations. Technically, SAF is not solving the computation issue that exists in SAM, because it completely discards the basic idea of SAM. So it is somehow not well appropriate to directly compare with SAF in the current topic in terms of the computational efficiency. We have added discussions in the revised paper.
> >
> > Here, we have also discussed with some typical works that targets to improve the computational efficiency in SAM. Importantly, we would like to show that our RST does not conflict with these efficient methods, which means that we do not have to compare with these corresponding efficient techniques. We could easily notice that when selecting performing SAM algorithm in RST, we would not impose any operation on SAM algorithm. In fact, we could adopt these efficient methods when selecting SAM algorithm to further improve the computational efficiency of performing SAM algorithm. The following table only shows the results of ResNet18. And for detailed results, the reviewer could check the appendix B.6 in the newly uploaded version, including the results regarding VGG16BN and WideResNet28-10.
> >
> > LookSAM "Liu et al. 2022, Towards Efficient and Scalable Sharpness-Aware Minimization, CVPR2022"
> > |Method|Time|C-10 Error|C-100 Error|
> > |--|:--:|:--:|:--:|
> > |SGD|15.6|4.48|20.79|
> > |LookSAM|+5.6|4.06|20.30|
> > |LookSAM + G-RST[50%]|+3.0|4.18|20.44|
> > |LookSAM + G-RST[75%]|+4.4|3.94|20.11|
> >
> > ESAM "Du et al. 2022, Efficient Sharpness-aware Minimization for Improved Training of Neural Networks, ICLR2022"
> > |Method|Time|C-10 Error|C-100 Error|
> > |--|:--:|:--:|:--:|
> > |SGD|24.4|4.66|20.98|
> > |ESAM|+18.6|4.05|20.28|
> > |ESAM + G-RST[50%]|+9.5|4.41|20.72|
> > |ESAM + G-RST[75%]|+14.0|4.08|20.21|
> >
> > SSAM "Mi et al. 2022, Make Sharpness-Aware Minimization Stronger: A Sparsified Perturbation Approach, Nips2022"
> > |Method|Time|C-10 Error|C-100 Error|
> > |--|:--:|:--:|:--:|
> > |SGD|24.4|4.66|20.98|
> > |SSAM|+21.0|3.89|20.17|
> > |SSAM + G-RST[50%]|+10.7|4.03|20.41|
> > |SSAM + G-RST[75%]|+15.8|3.83|20.19|
> >
> > It should be pointed that here, ESAM and SSAM are both implemented on the git repository https://github.com/Mi-Peng/Sparse-Sharpness-Aware-Minimization, where one A100 GPU is used.As we could see in the table, for all these efficient techniques, our RST could naturally improve the computational efficiency further since all these methods could have the choice to selecting performing base learning algorithm. However, if the selecting probability in RST is relatively low (0.5 in the table), it may harm the mixing effect, where the models may not be well trained. On the other hand, as properly raising the selecting probability (0.75 in the table), it is possible to acquire comparable results with these efficient techniques. We thank again for the review time and valuable comment, and the reviewer could check Appendix for more details.
> >
> > `Q: Also, I was confused that why the selected architectures are different on CIFAR and ImageNet dataset?`
> >
> > A: We would like to thank the reviewer for the valuable comment. Actually, the validation here strictly follows the convention in the typical contemporary literatures regarding SAM related works (such as SAM, ESAM, ASAM, gradient norm regularization etc.). As the reviewer could see in the related papers, they all use different architectures for Cifar and ImageNet. We suppose this may partly because that some network architectures are not well suitable for training on ImageNet because of the computation cost and the similarity with ResNet architectures.
> >
> > We thank the reviewer for the valuable time and comments.

---

### Official Review · Reviewer_hEbB · 2022-10-25

**Confidence:** 4
**Clarity, Quality, Novelty And Reproducibility:** Please see the weakness part.
**Correctness:** 3
**Technical Novelty And Significance:** 2
**Empirical Novelty And Significance:** 2
**Recommendation:** 5

**Strength And Weaknesses:**

Strength:
- The proposed RST can reduce the extra computational cost of the vanilla SAM from 1 to $\sum p(t) / T$ in average, and preserves similarly good performances.
- The paper is easy to follow.

Weakness:
- The novelty of this paper is my only concern. The idea of applying SAM to a subset of parameters or iterations has been explored in several papers [Mi et al. 2022, Liu et al. 2022, Du et al. 2022]. For example, Liu et al. 2022 propose to only periodically calculate the inner gradient ascent across the training iterations; Liu et al. 2022 and Du et al. 2022 propose to select a subset of parameters to calculate the inner gradients in each iterations. Over expectations, randomly selecting a subset of parameters in iterations is same as randomly select interactions. And the later can be considered as a special case of the former --- alternatively masking out all parameters.


Mi et al. 2022, Make Sharpness-Aware Minimization Stronger: A Sparsified Perturbation Approach

Du et al. 2022, Efficient Sharpness-aware Minimization for Improved Training of Neural Networks

Liu et al. 2022, Towards Efficient and Scalable Sharpness-Aware Minimization

**Summary Of The Paper:**

Sharpness-aware minimisation (SAM) has been shown powerful to train high-performance deep learning models, but it also incurs at least double computational cost due to the extra back-propagation for the sharpness estimation. To improve the efficiency of the vanilla SAM, this paper proposes an training scheme, dubbed Randomized Sharpness-Aware Training (RST). RST performs a Bernoulli trial at each iteration to choose randomly from base algorithms (SGD) and SAM. The probabilities of Bernoulli trials at each time are  determined by a predefined scheduling function $p(t)$. The average extra time is reduced from 1 to $\sum p(t) / T$.

**Summary Of The Review:**

The paper is easy to follow. The empirical experiments are enough to support the arguments. However, the novelty of this paper is the concern. The idea of selecting a subset of parameters or iterations for implementing SAM has been explored in several recent works.

---

> ### Author Response · Authors · 2022-11-12
> **We thank the reviewer for all the valuable and constructive comments!**
>
> `Q: The novelty of this paper is my only concern. The idea of applying SAM to a subset of parameters or iterations has been explored in several papers [Mi et al. 2022, Liu et al. 2022, Du et al. 2022]. For example, Liu et al. 2022 propose to only periodically calculate the inner gradient ascent across the training iterations; Liu et al. 2022 and Du et al. 2022 propose to select a subset of parameters to calculate the inner gradients in each iterations. Over expectations, randomly selecting a subset of parameters in iterations is same as randomly select interactions. And the later can be considered as a special case of the former --- alternatively masking out all parameters.`
>
>
>
> We would like to thank the reviewer for the valuable comment. Firstly, we would like to discuss the difference between the weight masking method and our randomized policy. We understand the reviewer's point in regards to the mentioned "expectation" intuition. However, the two strategies have very different effect in terms of the computational efficiency. The reviewer could find that in ESAM paper, the ESAM's authors have explained why the improvement of computation efficiency is limited when using such masking strategy. In a word, due to the chain rule in BP, computing the gradients of weights in shallower layers requires computing the gradients in deeper layers. Therefore, gradients of many masked-out weights still need to be computed despite the fact that they have been masked out. Further, the reviewer could also find that the empirical results in ESAM confirm this demonstrations, where the computational efficiency is improved by only 10\% even the majority of the weights are not selected. Meanwhile, Sparse SAM follows the very similar idea of masking weights as ESAM. Although the Sparse SAM's authors show that their method theoretically requires much less FLOPs, yet still due to the impact of the chain rule in BP in practice, the actual training wall time is pretty close to the standard SAM training, even if we have set a very high sparse rate in Sparse SAM. The reviewer could give it a try https://github.com/Mi-Peng/Sparse-Sharpness-Aware-Minimization, and we also have uploaded the corresponding training logs in our repo for your reference. Besides, from the perspective of practical implementations, randomly selecting to records the gradients is not quite friendly to current deep learning frameworks because we may frequently alter the computational graph. For JAX, this may probability cause a re-complication in auto-differential framework, which could instead largely burden the computational efficiency. As for our method, we do not need to worry about all these concerns. Besides, we have also provided a comprehensive analysis to show how and why our method works.
>
> Next, more importantly, we would like to show that our RST does not conflict with these efficient methods, which means that we do not have to compare with these corresponding efficient techniques. We could easily notice that when selecting performing SAM algorithm in RST, we would not impose any operation on SAM algorithm. In fact, we could adopt these efficient methods when selecting SAM algorithm to further improve the computational efficiency of performing SAM algorithm. The following table only shows the results of ResNet18. And for detailed results, the reviewer could check the appendix B.6 in the newly uploaded version, including the results regarding VGG16BN and WideResNet28-10.
>
> LookSAM
> |Method|Time|C-10 Error|C-100 Error|
> |--|:--:|:--:|:--:|
> |SGD|15.6|4.48|20.79|
> |LookSAM|+5.6|4.06|20.30|
> |LookSAM + G-RST[50%]|+3.0|4.18|20.44|
> |LookSAM + G-RST[75%]|+4.4|3.94|20.11|
>
> ESAM
> |Method|Time|C-10 Error|C-100 Error|
> |--|:--:|:--:|:--:|
> |SGD|24.4|4.66|20.98|
> |ESAM|+18.6|4.05|20.28|
> |ESAM + G-RST[50%]|+9.5|4.41|20.72|
> |ESAM + G-RST[75%]|+14.0|4.08|20.21|
>
> SSAM
> |Method|Time|C-10 Error|C-100 Error|
> |--|:--:|:--:|:--:|
> |SGD|24.4|4.66|20.98|
> |SSAM|+21.0|3.89|20.17|
> |SSAM + G-RST[50%]|+10.7|4.03|20.41|
> |SSAM + G-RST[75%]|+15.8|3.83|20.19|
>
> It should be pointed that here, ESAM and SSAM are both implemented on the git repository https://github.com/Mi-Peng/Sparse-Sharpness-Aware-Minimization, where one A100 GPU is used.As we could see in the table, for all these efficient techniques, our RST could naturally improve the computational efficiency further since all these methods could have the choice to selecting performing base learning algorithm. However, if the selecting probability in RST is relatively low (0.5 in the table), it may harm the mixing effect, where the models may not be well trained. On the other hand, as properly raising the selecting probability (0.75 in the table), it is possible to acquire comparable results with these efficient techniques.
>
> Besides, we have also added extra results of our RST on SAM variants like ASAM and GSAM at Appendix B.5. The reviewer could check if interested. We thank the reviewer for the valuable time and comments.

---

### Official Review · Reviewer_eiyQ · 2022-10-27

**Confidence:** 3
**Correctness:** 3
**Technical Novelty And Significance:** 3
**Empirical Novelty And Significance:** 2
**Recommendation:** 5

**Clarity, Quality, Novelty And Reproducibility:**

This paper is well-written, and it is easy to follow the contents.

There is an implementation code, and it is easy to reproduce.

**Strength And Weaknesses:**

Strength points
1. This paper aims to improve the efficiency of SAM, the important problems.
2. This paper raises interesting ideas such as randomized sharpness and its general extension G-RST that adjust regularization degree freely for any scheduling function.

Weakness points
1. The concept of this paper is similar to the other papers [1]. This paper should cite and discuss the difference between this paper and another paper.
2. This paper is necessary to include diverse related works and efficient computation of SAM. For example, there are three research works, [2], [3], [4]. This paper only cites and discusses the [4]. However, there is no experimental comparison with [4]. I suggest that this paper should discuss and compare the performance with [2], [3], [4].

[1] Zhao, Yang, Hao Zhang, and Xiuyuan Hu. "SS-SAM: Stochastic Scheduled Sharpness-Aware Minimization for Efficiently Training Deep Neural Networks." arXiv preprint arXiv:2203.09962 (2022).
[2] Du, Jiawei, et al. "Sharpness-Aware Training for Free." arXiv preprint arXiv:2205.14083 (2022).
[3] Liu, Yong, et al. "Towards efficient and scalable sharpness-aware minimization." Proceedings of the IEEE/CVF Conference on Computer Vision and Pattern Recognition. 2022.
[4] Du, Jiawei, et al. "Efficient sharpness-aware minimization for improved training of neural networks." arXiv preprint arXiv:2110.03141 (2021).

**Summary Of The Paper:**

Recently, sharpness-aware training such as SAM has drawn large attention, because of its provably guarantee its significant performance improvement. However, SAM requires a huge additional computation cost, and it is not easy to adopt SAM on a large-scale model or a real-time analysis system. This paper provides an efficient computation for SAM, RST, and G-RST. They adopt randomized sharpness-aware training. The idea is simple, and it works well.

**Summary Of The Review:**

This paper raises an interesting concept, but similar ideas were already suggested in other venues.

Discussion and additional experimental comparisons are necessary.

---

> ### Author Response · Authors · 2022-11-12
> **We thank the reviewer for the valuable and constructive comments!**
>
> `Q: The concept of this paper is similar to the other papers [1]. This paper should cite and discuss the difference between this paper and another paper.`
>
> We thank the reviewer for the helpful comment. This mentioned paper is obviously an unfinished work. And here we publicly promise no violation of terms in regards to plagiarism and ethics.
>
> `Q: This paper is necessary to include diverse related works and efficient computation of SAM. For example, there are three research works, [2], [3], [4]. This paper only cites and discusses the [4]. However, there is no experimental comparison with [4]. I suggest that this paper should discuss and compare the performance with [2], [3], [4].`
>
> We thank the reviewer for the valuable comment. We have discussed all these methods in the newly uploaded paper. And importantly, we would like to show that our RST does not conflict with these efficient methods, which means that we do not have to compare with these corresponding efficient techniques. We could easily notice that when selecting performing SAM algorithm in RST, we would not impose any operation on SAM algorithm. In fact, we could adopt these efficient methods when selecting SAM algorithm to further improve the computational efficiency of performing SAM algorithm. The following table only shows the results of ResNet18. And for detailed results, the reviewer could check the appendix B.6 in the newly uploaded version, including the results regarding VGG16BN and WideResNet28-10.
>
> LookSAM[3]
> |Method|Time|C-10 Error|C-100 Error|
> |--|:--:|:--:|:--:|
> |SGD|15.6|4.48|20.79|
> |LookSAM|+5.6|4.06|20.30|
> |LookSAM + G-RST[50%]|+3.0|4.18|20.44|
> |LookSAM + G-RST[75%]|+4.4|3.94|20.11|
>
> ESAM[4]
> |Method|Time|C-10 Error|C-100 Error|
> |--|:--:|:--:|:--:|
> |SGD|24.4|4.66|20.98|
> |ESAM|+18.6|4.05|20.28|
> |ESAM + G-RST[50%]|+9.5|4.41|20.72|
> |ESAM + G-RST[75%]|+14.0|4.08|20.21|
>
> SSAM
> |Method|Time|C-10 Error|C-100 Error|
> |--|:--:|:--:|:--:|
> |SGD|24.4|4.66|20.98|
> |SSAM|+21.0|3.89|20.17|
> |SSAM + G-RST[50%]|+10.7|4.03|20.41|
> |SSAM + G-RST[75%]|+15.8|3.83|20.19|
>
> It should be pointed that here, ESAM and SSAM are both implemented on the git repository https://github.com/Mi-Peng/Sparse-Sharpness-Aware-Minimization, where one A100 GPU is used. As we could see in the table, for all these efficient techniques, our RST could naturally improve the computational efficiency further since all these methods could have the choice to selecting performing base learning algorithm. However, if the selecting probability in RST is relatively low (0.5 in the table), it may harm the mixing effect, where the models may not be well trained. On the other hand, as properly raising the selecting probability (0.75 in the table), it is possible to acquire comparable results with these efficient techniques. We thank again for the review time and valuable comment, and the reviewer could check Appendix for more details.
>
> Besides, we would like to particularly discuss the paper "Sharpness-Aware Training for Free" (SAF) [2], Nips 2022. SAF proposes to minimize the KL-divergence between the output distributions yielded by the current model and the moving average of past models. It should be particularly noted that unlike SAM, SAF follows another way, where no explicit regularization is imposed within the neighborhood region during the whole training process. In fact, this makes SAF seem more like the idea of knowledge distillation than sharpness-aware learning, where the teacher model is the moving average of past models and the student model is the current model. Besides, just as the results reported in their paper, the superiority of SAF over SAM variants still requires further demonstrations. Technically, SAF is not solving the computation issue that exists in SAM, because it completely discards the basic idea of SAM. So it is somehow not well appropriate to directly compare with SAF in the current topic in terms of the computational efficiency. We have added discussions in the revised paper.
>
> We thank the reviewer for the valuable time and comments.

---

### Official Review · Reviewer_cVKR · 2022-11-01

**Confidence:** 4
**Correctness:** 4
**Technical Novelty And Significance:** 3
**Empirical Novelty And Significance:** 3
**Recommendation:** 8

**Clarity, Quality, Novelty And Reproducibility:**

This paper is very well written overall. The paper is also original to some extent in the sense that although there exist SAM variants attempting to improve on SAM’s computational aspect, unlike most of these works this work develops based on matured techniques and optimisation characteristics and provides very well thought-out and reliable results.


**Strength And Weaknesses:**

- The paper is written in a very clear and professional way.
- The paper is very solid with balanced views and analysis.
- The paper provides convergence analysis which somehow lacks in SAM literature.
- The resulting algorithm is simple and effective.
- I literally didn’t find any flaws in the paper but thought adding more experiments on large scale and different domains could make the paper even stronger.


**Summary Of The Paper:**

This paper presents new training methods called RST and G-RST to extend the geometry inspired training method SAM and improve its computational efficiency.
Based on randomized gradient boosting RST randomly selects between SGD and SAM with probability based on parameterized Bernoulli distribution.
The authors explore different parameterization schemes for such a scheduling function and analyze their effect on computations and performance trade-off.
The authors also develop RST’s convergence properties for non-convex stochastic cases where the classes of objective functions are smooth and strongly convex.
The paper evaluates RST and G-RST on multiple image classification tasks showing that the proposed methods can save 50% computations while performing on-par or even better than the original SAM.


**Summary Of The Review:**

Highly recommended for interested readers on SAM literature.

---

> ### Author Response · Authors · 2022-11-12
> **We thank the reviewer for the valuable and constructive comments!**
>
> We thank the reviewer for the valuable and constructive comments. We are very gratified that the reviewer admits our work. We totally agree with the reviewer's point that adding diverse experimental results could complete our work. Therefore, we start from trying to check our method on the noisy-label experiments. The following table shows the results of ResNet18 trained with different percentage of label corruptions on Cifar10. We could see that using G-RST could give comparable effect with SAM in the tasks of label corruption. And we would add these results and more results in the final paper.
>
> |Method|Corruption|C-10 Error|
> |--|:--:|:--:|
> |SGD|0.2|8.84|
> |SAM|0.2|7.25|
> |G-RST|0.2|7.10|
>
> |Method|Corruption|C-10 Error|
> |--|:--:|:--:|
> |SGD|0.4|12.28|
> |SAM|0.4|9.57|
> |G-RST|0.4|9.66|
>
> |Method|Corruption|C-10 Error|
> |--|:--:|:--:|
> |SGD|0.6|17.64|
> |SAM|0.6|13.59|
> |G-RST|0.6|13.23|
>
> |Method|Corruption|C-10 Error|
> |--|:--:|:--:|
> |SGD|0.8|32.09|
> |SAM|0.8|30.81|
> |G-RST|0.8|30.94|
>
> Besides, we have additionally performed detailed experimental results regarding mixing with other efficient techniques and SAM variants in the Appendix to further show the effectiveness of our method. The reviewer could check if interested. Again thanks for the valuable time and comments.

---

### Author Response · Authors · 2022-11-12
**A description to the revised paper.**

We thank all the reviewers for their constructive and valuable comments. We have carefully investigated all the comments one by one. According to these comments, we have made many improvements to the work and have uploaded the revised paper.

The core or maybe the only concern could be summarized as the comparisons with other recent new contemporary works that target to improve the computational efficiency in SAM. We have added discussions in the Introduction. And we have provided a detailed investigation to show that our RST does not conflict with these efficient methods. Appendix B.6 (Page 18-20) shows the mixing effect of our method with other related techniques.

Additionally, we have further investigated the effectiveness of our RST on SAM variants. Appendix B.5 shows the corresponding results of our RST on ASAM and GSAM.

We thank the reviewer for their review time.

---

### Comment · Area_Chair_jnGS · 2022-11-18
**Please discuss**

Dear reviewers,

Since there is a rather large variance among the scores, I would appreciate if you can takes a look at the authors' responses and the other reviews to check if you would like to update your score and to engage with the authors.

Best,
Your AC

---

### Decision · Program_Chairs · 2023-01-20

**Decision:**

Reject

**Justification For Why Not Higher Score:**

I have explained this above.

**Justification For Why Not Lower Score:**

I cannot go any lower.

**Metareview: Summary, Strengths And Weaknesses:**

The main contribution of this paper is to accelerate SAM. This is a worthy endeavor.

This paper was perceived to be borderline and so as the AC, I spent a significant amount of time discussing it with the reviewers and looking through the paper. Unfortunately, my recommendation is a reject. This is because the main contribution is very similar to LookSAM (Liu et al., 2022), which is to do SAM once every x iterations. The remainder of the paper is to explain why the method works theoretically; which is good.

I understand that it is impossible to compare the method to recently proposed variants of SAM (which we published a few days before the ICLR deadline). However, the main idea overlaps significantly with LookSAM, so this has an adverse effect on the novelty of the whole contribution.

**Summary Of Ac-Reviewer Meeting:**

Yes. I did have email discussions with the reviewers. In particular, the reviewer who was an outlier did maintain that there was some novelty in the method and theoretical analyses. I do not disagree. However, I maintain that the overlap with the existing works (especially LookSAM) results in the paper not passing the bar for ICLR.